

# A comprehensive study on hygroscopic behaviour and nitrate depletion of NaNO₃ and dicarboxylic acid mixtures: Implication for the influence factors of nitrate depletion

Shuaishuai Ma[a], Qiong Li[b], Yunhong Zhang[a]

[a] *The Institute of Chemical Physics, School of Chemistry and Chemical Engineering, Beijing Institute of Technology, Beijing 100081, PR China*

[b] *Shanghai Key Laboratory of Atmospheric Particle Pollution and Prevention, Department of Environmental Science & Engineering, Institute of Atmospheric Sciences, Fudan University, Shanghai 200433, PR China*

*Correspondence to*: Yunhong Zhang (yhz@bit.edu.cn)

**Abstract.** The nitrate depletion and $HNO_3$ release in internally mixed nitrate and dicarboxylic acids (DCAs) particles have been widely detected in field and laboratory studies. Nevertheless, considerable discrepancies are still present among these measurements, and the influence factors for this acid-displacement reaction have not yet been elucidated. In this work, the hygroscopic growth and chemical composition evolution of mixtures of $NaNO_3$ and DCAs, i.e., oxalic acid (OA), malonic acid (MA), and glutaric acid (GA), were measured using attenuated total reflectance Fourier transform infrared spectroscopy (ATR-FTIR) and vacuum FTIR techniques. The $HNO_3$ release from $NaNO_3$/OA mixtures was observed in both the measurements, owing to the relatively high acidity of OA. At the same time, the $NaNO_3$ phase state was found to act as a key regulator of nitrate depletion. Amorphous $NaNO_3$ solids at relative humidity (RH) < 5% were inert to liquid OA. With increasing RH, the mixtures experienced three interesting stages of phase changes showing different $HNO_3$ release rates, e.g., at around 15% RH, the slow $HNO_3$ release was detected by the vacuum IR spectra, potentially indicating the transformation of amorphous solids to semisolid $NaNO_3$; in the second stage (sudden RH increase from ~ 15% to 61%), the $HNO_3$ release rate was increased by about an order of magnitude; when $NaNO_3$ deliquescence occurred in the third stage, this displacement reaction proceeded due to more available $NO_3^-$ ions formation. Compared to OA, MA and GA reacted with nitrate only in vacuum FTIR measurement, while in ATR-FTIR measurement, the mixtures tended to be effloresced completely without nitrate depletion. Further, the influences of ambient pressure, chemical composition, and water activity on $HNO_3$ release rates were estimated via Maxwell steady-state diffusive mass transfer equation. The results showed that weaker acidity of MA and GA as well as relatively lower $HNO_3$ diffusion rate in ambient gas phase mainly contributed to the unobserved nitrate depletion in ATR-FTIR measurement. Our findings reveal that chemical component, phase state, and water activity of particles, as well as $HNO_3$ gas phase diffusion play crucial roles on $HNO_3$ release from nitrate and DCAs mixtures. This work may provide a new perspective on nitrate depletion in the aging processes during transport of atmospheric aerosols.



## 1 Introduction

Aerosol particles in the atmosphere can play crucial roles in determining Earth's climate, air quality, and human health (Pöschl, 2006; Stevens and Feingold, 2009; Carslaw et al., 2013; Ramanathan et al., 2001; Brown et al., 2006), depending on their various physicochemical properties, e.g., chemical composition, phase state, volatility, reactivity, hygroscopicity, the ability to absorb and scatter solar light as well as act as cloud condensation nuclei (CCN) (Pöschl, 2006; Mcfiggans et al., 2006; Haywood and Boucher, 2000; Farmer et al., 2015; Shiraiwa et al., 2017; Freney et al., 2018; Kuwata and Martin, 2012).

Sea salt and mineral dust aerosols can provide highly reactive surfaces for nitrates production through heterogeneous reactions of gaseous nitrogen oxides such as $HNO_3$, $N_2O_5$, $NO_2$, and $NO_3$ (Gibson et al., 2006; Song and Carmichael, 2001; Finlayson-Pitts and Hemminger, 2000). Furthermore, heterogeneous and aqueous oxidation of DCA precursors and gas-particle partitioning of DCAs in the atmosphere will cause the internal mixing of DCAs with sea salt and mineral dust aerosols (Tervahattu et al., 2002; Sullivan and Prather, 2007; Laskin et al., 2012; Wang et al., 2010), greatly influencing the hygroscopic behaviour and surface tension of mixed particles (Facchini et al., 1999; Yu, 2000; Jing et al., 2018).

It is well known that the displacement of strong acids, i.e., HCl or $HNO_3$, by weak organic acids, e.g., DCAs, is not thermodynamically favoured in bulk solutions. Whereas, the nitrate and chloride depletion in mixed nitrate/chloride and organic acid particles has been widely detected in field and laboratory measurements, which could be expressed as (Laskin et al., 2012; Wang and Laskin, 2014; Ma et al., 2019b; Ghorai et al., 2014; Shao et al., 2018)

$$NaCl/NaNO_3 (aq) + HA (org)(aq) \leftrightarrow NaA (org)(aq,s) + HCl/HNO_3 (g)(\uparrow) \tag{R1}$$

The driving force for this displacement reaction is mainly governed by the acidity difference and volatility difference between the organic acids and $HCl/HNO_3$ (Laskin et al., 2012; Wang and Laskin, 2014; Chen et al., 2021). The acidity difference tends to shift the reaction equilibrium to the left, demonstrating the more dominating driving force for the substitution of strong acids by weak organic acids is the volatility difference (Laskin et al., 2012; Chen et al., 2021). For instance, dissociated HCl concentration is $\sim 10^{10}$ times higher than dissociated citric acid within mixed NaCl/citric acid (1/1 molar ratio) droplets, while the equilibrium gas phase concentration of HCl far exceeds that of citric acid with a factor of $\sim 10^{19}$, suggesting the HCl partition into the gas phase would manage the direction of acid-displacement reaction (Laskin et al., 2012). In general, the less acidity difference, i.e., higher acid dissociation constants ($K_{a1}$) of organic acids, and larger volatility difference, i.e., higher Henry's law constants ($K_H$) of organic acids, are favourable for gaseous $HCl/HNO_3$ liberation.

As indicated by previous studies, reactions between nitrate or chloride and OA, which had a log ionization constant ($pK_{a1}$) of 1.23 (Haynes and Lide, 2011), always occurred in mixed aerosols, causing the formation of less hygroscopic metal oxalates (Ma et al., 2013, 2019a, b; Ma and He, 2012). Nevertheless, there were considerable discrepancies among earlier observations for internally mixed NaCl and MA ($pK_{a1} = 2.83$). Laskin et al. (2012), Ghorai et al. (2014), Laskina et al. (2015), and Li et al. (2017) supported the occurrence of HCl substitution by MA, while Choi and Chan (2002), Pope et al.


(2010), and Ma et al. (2013) did not observe this displacement reaction. Likewise, succinic acid (SA) ($pK_{a1}$ = 4.20) was suggested not to react with internally and externally mixed NaCl by Ma et al. (2013), while Ghorai et al. (2014) found that

GA ($pK_1$ = 4.32) was reactive to NaCl. In another publication, nitrates were proved to be reactive to MA and GA leading to nitrate depletion (Wang and Laskin, 2014). Indeed, the differences in particle sizes, substrate materials, acidity and volatility of DCAs may be responsible for these controversial results, but it is true that the interactions between organic acids and nitrate/chloride are still not clear.

To the best of our knowledge, the influence of particle-phase state on nitrate depletion has not been considered yet. As

Wang et al. (2017) indicated, the substrate supporting OA droplets would crystallize to form OA dihydrate at ~ 71% RH, which further lost crystalline water to form anhydrous OA at ~ 5% RH; meanwhile, no deliquescence was observed upon hydration. Pure $NaNO_3$ droplets might not be effloresced but convert into amorphous state at low RH (Hoffman et al., 2004; Liu et al., 2008), or effloresced at certain RH values (Lamb et al., 1996; Zhang et al., 2014). Furthermore, atmospheric aerosols can exist in semisolid, glassy, highly viscous, or solid-liquid mixing states (Krieger et al., 2012; Mikhailov et al.,

2009; Virtanen et al., 2010), depending on varying ambient RH, temperature, heterogeneous inclusions and so on (Ma et al., 2021). Additionally, the particle-phase state has been proved to play a critical role in determining the reactivity of secondary organic material (SOM) upon ammonia exposure (Kuwata and Martin, 2012). These scenarios pose a key issue concerning how the particle-phase state would affect nitrate depletion in sea salt and mineral dust particles.

In this work, two measurement techniques, i.e., ATR-FTIR and vacuum FTIR, were carried out to measure the

hygroscopic behaviour and nitrate depletion of mixed $NaNO_3$ and DCAs particles. The effects of $HNO_3$ gas phase diffusion, as well as chemical component, phase state, and water activity of mixed particles on nitrate depletion were further explored. This work would enhance our understanding of interaction mechanisms of DCAs and nitrate.

## 2 Experimental Section

### 2.1 Sample preparation

The 0.1 mol $L^{-1}$ mixture solutions of $NaNO_3$/OA, $NaNO_3$/MA, and $NaNO_3$/GA with molar ratio of 1:1 (or 3:1) were prepared by dissolving nitrate and DCAs into ultrapure water (18.2 MΩ cm resistivity). The bulk solutions were nebulized ultrasonically to produce aerosol droplets deposited on two $CaF_2$ windows in vacuum FTIR measurement and the Ge substrate in ATR-FTIR measurement.

### 2.2 Vacuum FTIR measurement

The vacuum FTIR technique was composed of a vacuum FTIR spectrometer and a RH controlling system. The experimental apparatus and method have been described in detail by our previous studies (Leng et al., 2015; Zhang et al., 2017; Ma et al., 2019c), thus a brief was introduced here. The vacuum FTIR spectrometer (Bruker VERTEX 80v) consisted of a vacuum optics bench, a sample compartment and a vacuum pump. The RH controlling system was composed of a high purity water





reservoir, a sample chamber, and another vacuum pump. Water vapor from the water reservoir was fed into the sample

chamber, and was pumped out by the vacuum pump. Two solenoid valves were installed in the pipeline of water vapor to instantaneously switch water vapor inlet and outlet. Meanwhile, two needle valves were configured to respectively adjust the flow rates of water vapor inlet and outlet.

The radius of deposited droplets on $CaF_2$ windows was in the range of ~ 1-3 μm (Zhang et al., 2017). After the nebulization, the $CaF_2$ windows were installed onto the sample chamber to seal it. The air in optics bench, sample

compartment and sample chamber was pumped out to remove water vapor and $CO_2$. The baseline pressure in optics bench and sample compartment was pumped to ~ 0.21 kPa and the sample chamber arrived to ~ 0.01 kPa to remove water vapor and $CO_2$. A differential pressure transmitter (Rosemount 3051, accuracy > 0.5%) was used to measure water vapor pressure in the sample chamber, which could be used to calibrate the ambient RH (Zhang et al., 2017). The water content of deposited particles could be estimated by the integrated absorbance of stretching vibration band of liquid water molecules at 3400 $cm^{-1}$

(Ma et al., 2019c). The resolution of collected IR spectra was 4 $cm^{-1}$. All measurements were made at 23-26 ℃.

### 2.3 ATR-FTIR measurement

Detailed description of ATR-FTIR measurement has been reported elsewhere (Zhang et al., 2014; Ren et al., 2016). Briefly, the IR spectra of deposited particles on a horizontal ATR (Spectra-Tech Inc. USA) accessory with the Ge substrate were measured by a Nicolet Magna-IR model 560 FTIR spectrometer equipped with a liquid-nitrogen cooled mercury-cadmium-

telluride (MCT) detector. The ambient RH in the sample cell was controlled by adjusting the flow rate ratio of gas streams of dry and humidified nitrogen with a total flow rate of 800 mL·$min^{-1}$. The ambient RH and temperature were recorded by a hygrometer (Centertek Center 310, accuracy of ± 2.5%) in the outlet of sample cell. The diameter of deposited droplets was 1-5 μm with a median diameter of ~ 3 μm (Zhang et al., 2014). The IR spectra were collected between 4000 and 600 $cm^{-1}$ with a resolution of 4 $cm^{-1}$. Similar to vacuum FTIR measurement, the water content in aerosol particles was determined by

the integrated absorbance of the stretching vibration band of liquid water.





## 3 Results and Discussions

### 3.1 Hygroscopic growth and chemical composition evolution of NaNO₃/OA mixtures

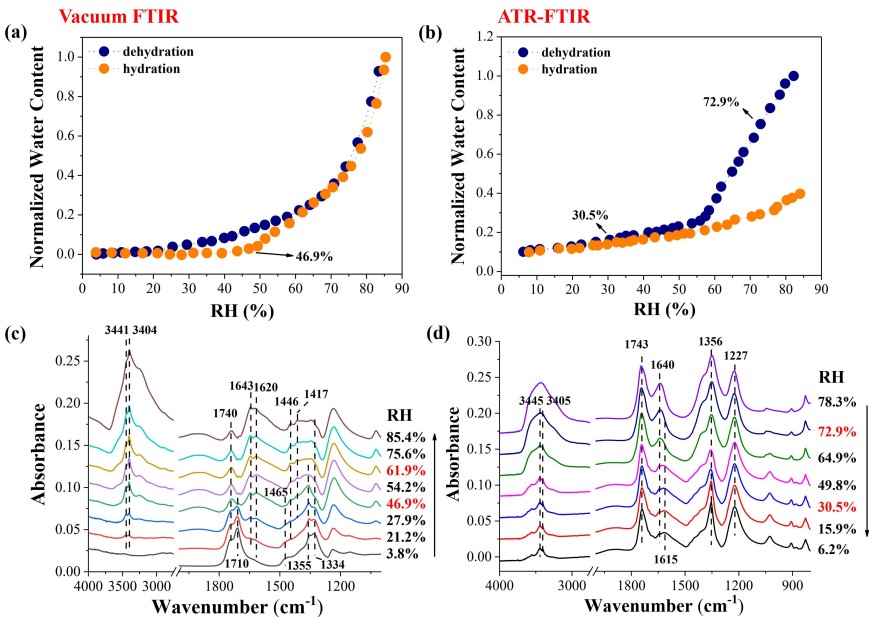

**Figure 1.** Hygroscopic growth curves of 1:1 mixed NaNO₃/OA particles measured by vacuum FTIR (a) and ATR-FTIR (b), as well as
corresponding IR spectra during the humidification in vacuum FTIR measurement (c) and during the dehumidification in ATR-FTIR
measurement (d).

The hygroscopic behaviour and IR features of individual components, i.e., NaNO₃, OA, MA, and GA, are shown in the
Supplement. Figure 1 displays the hygroscopic growth and chemical composition evolution of 1:1 NaNO₃/OA mixtures
during a RH cycle. In vacuum FTIR measurement, the deposited droplets on CaF₂ windows are first dried in vacuum, and
then undergo a humidification-dehumidification cycle. As shown in Fig. 1c, there are two feature bands assigned to the
stretching mode of COOH functional groups ($\nu$(COOH)) at ~ 3.8% RH, i.e., the 1740 cm⁻¹ band assigned to liquid OA and
1710 cm⁻¹ band assigned to anhydrous OA (Wang et al., 2019), indicating the coexistence of liquid OA and anhydrous OA.
The 1355 cm⁻¹ band is attributed to NO₃⁻ asymmetric stretching vibration ($\nu_3$(NO₃⁻)) of amorphous NaNO₃ solids, as
discussed in the Supplement. The weak absorption at 1620 cm⁻¹ indicates small amounts of crystalline oxalate formation
(Hind et al., 1998; Wang et al., 2019), judged from the IR features of Na₂C₂O₄ solids shown in Fig. S4. In addition, the peak
at 1465 cm⁻¹ is attributed to O-H bending mode of HC₂O₄⁻ ions (Villepin and Novak, 1971), indicative of the NaHC₂O₄



formation. Likewise, Wang et al. (2017) observed the formation of $NH_4HC_2O_4$ in mixed $(NH_4)_2SO_4$/OA droplets upon drying. These scenarios confirm the nitrate depletion and $HNO_3$ release from $NaNO_3$/OA mixtures in the vacuuming process. As RH increases to 21.2%, the feature bands at 3441 and 3404 cm$^{-1}$ appear, indicating the transformation of anhydrous OA

to OA dihydrate. After that, the stronger 1620 cm$^{-1}$ and 1417 cm$^{-1}$ bands, assigned to $C_2O_4^{2-}$ ion vibrating (Wang et al., 2019), and weaker 1465 cm$^{-1}$ band indicate the conversion of aqueous $NaHC_2O_4$ to crystalline $Na_2C_2O_4$. Thus, the acid-displacement reaction for $NaNO_3$/OA mixtures can be expressed as

$$HC_2O_4^-(aq) + H^+(aq) + 2Na^+(aq) + 2NO_3^-(aq) \rightarrow HC_2O_4^-(aq) + 2Na^+(aq) + NO_3^-(aq) + HNO_3\ (g)(\uparrow) \rightarrow C_2O_4^{2-}(aq) +$$
$$H^+(aq) + 2Na^+(aq) + NO_3^-(aq) \rightarrow Na_2C_2O_4(s) + HNO_3(g)(\uparrow) \tag{R2}$$

In previous studies, the reaction between NaCl or $NaNO_3$ and OA was found to produce disodium oxalate (Ma et al., 2013; Ma et al., 2019b). To our knowledge, the formation of intermediate product of sodium hydrogen oxalate is first observed here.

As shown in Fig. 1a and 1c, we can infer that the $NaNO_3$ deliquescence proceeds at 46.9%-61.9% RH, significantly lower than that of pure $NaNO_3$ particles (seen Fig. S2). Likewise, Ma et al. (2013) found that the DRH of NaCl component

decreased in both external and internal NaCl/MA mixtures. Note that the OA dihydrate and crystalline $Na_2C_2O_4$ cannot be deliquesced due to their very high DRHs (Ma et al., 2013; Wang et al., 2019). The IR spectra during the dehumidification are shown in Fig. S5a. As seen, the 1357 cm$^{-1}$ band becomes sharper at ~ 13.6% RH, indicating the remaining $NaNO_3$ efflorescence. Moreover, the residue of reactants indicates the incomplete reaction between OA and $NaNO_3$, consistent with the observations by Ma et al. (2019b) for $Ca(NO_3)_2$/OA, $NaNO_3$/OA and $Zn(NO_3)_2$/OA mixed systems.

Figure 1b and 1d display the hygroscopic growth and IR spectra of 1:1 $NaNO_3$/OA mixtures measured by ATR-FTIR. Mixed particles undergo a dehumidification-humidification cycle. Upon dehydration, the water content of particles gradually decreases with decreasing RH. When the RH attains ~ 72.9%, the shoulder bands at 3445 and 3405 cm$^{-1}$ assigned to OA dihydrate appear, as shown in Fig. 1d. Meanwhile, a new peak located at 1615 cm$^{-1}$ is observed, suggesting the formation of disodium oxalate. As RH decreases to 30.5%, the 1356 cm$^{-1}$ band assigned to $v_3(NO_3^-)$ becomes sharper, indicating the

$NaNO_3$ efflorescence. Upon hydration, the water content at high RH is far below that upon dehydration, owing to the persistence of crystalline OA and $Na_2C_2O_4$ (Ma et al., 2013; Wang et al., 2017; Peng and Chan, 2001; Wu et al., 2011). Only $NaNO_3$ solids are deliquesced, identified by the broader $NO_3^-$ feature band at 1356 cm$^{-1}$ (seen Fig. S5b). For a better illustration of phase state changes of mixed particles, the optical images of 1:1 $NaNO_3$/OA mixture during two RH cycles are observed and shown in Fig. S6, and detailed discussion can be found in the Supplement.

Besides, as shown in Fig. 1c, the absorbance of 1620 cm$^{-1}$ band assigned to oxalate shows a slight increase at RH as low as 21.2%, implying the nitrate depletion proceeds at relatively low RH. Therefore, the effect of nitrate phase state on nitrate depletion needs to be further explored.



## 3.2 Effect of nitrate phase state on nitrate depletion for NaNO$_3$/OA mixtures

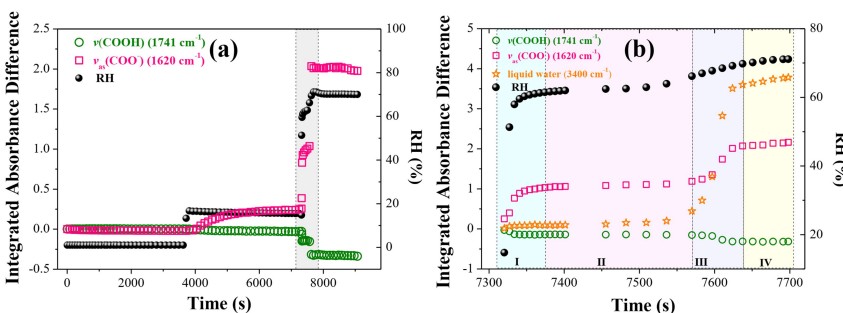

**Figure 2.** (a) Temporal changes in $\Delta A$ values of $\nu$(COOH) and $\nu_{as}$(COO$^-$) bands with stepwise increasing RH for 3:1 NaNO$_3$/OA mixed particles. (b) Temporal changes in $\Delta A$ values of $\nu$(COOH), $\nu_{as}$(COO$^-$) and liquid water bands with increasing RH corresponding to the shaded period in (a).

To better understand the phase state effect on this acid-displacement reaction, 3:1 NaNO$_3$/OA mixtures are dried in vacuum and then exposed to stepwise increasing RH. The integrated absorbance difference ($\Delta A$), derived from the deducted spectra of IR spectra at a certain time to that at the initial time, of 1741 cm$^{-1}$ band (assigned to COOH stretching mode, $\nu$(COOH)) and 1620 cm$^{-1}$ band (assigned to COO$^-$ asymmetric stretching vibration, $\nu_{as}$(COO$^-$)) as a function of time is determined and shown in Fig. 2a. First, the $\Delta A$ remains almost unchanged at RH < 5%. As RH increases to ~ 15%, the $\Delta A$ of $\nu_{as}$(COO$^-$) band exhibits a considerable increase, indicative of the occurrence of nitrate depletion. At the same time, the $\Delta A$ value of ~ 0.3 when the reaction equilibrium is reached implies that the nitrate depletion is limited at low RH. Note that the IR spectra changes with time at constant ~ 15% RH are supplied in Figure S7. Almost no 1620 cm$^{-1}$ band is observed at the initial time, suggesting almost no Na$_2$C$_2$O$_4$ formation in the vacuuming process, potentially owing to the minimization of HNO$_3$ release arises from rapid water evaporation (Ma et al., 2013). Then, the 1620 cm$^{-1}$ band appears and becomes stronger with time, suggesting Na$_2$C$_2$O$_4$ can be continuously produced at constant ~ 15% RH. Coupled with the particle morphology changes shown in Figure S6, we can infer the conversion of amorphous NaNO$_3$ solids to highly viscous semisolids due to the uptake of trace amounts of moisture. As RH continues to increase, the $\Delta A$ values of $\nu_{as}$(COO$^-$) and $\nu$(COOH) bands respectively increase or decrease. Finally, the $\Delta A$ values of the two bands remain constant with time.

Further, the variation in $\Delta A$ and water content as RH increases from ~ 14.8% to ~ 69.2% (shaded period in Fig. 2a) is processed and depicted in Fig. 2b. First, four stages are classed. In stage I, as RH increases from ~ 14.8% to ~ 61.0%, the liquid water content (yellow pentagram) shows a very slight increase, while the $\Delta A$ of $\nu_{as}$(COO$^-$) band (pink pane) and $\nu$(COOH) band (green circle) increases or decreases significantly. This implies that the absorbed moisture with increasing RH is favourable for nitrate depletion. In stage II, the RH increases slightly, meanwhile, the water content and the $\Delta A$ values remain almost unchanged. As the RH increases to ~ 66.2%, the deliquescence of remaining NaNO$_3$ occurs, causing a sharp



increase in liquid water content (stage III); meanwhile, the $\Delta A$ values of $\nu_{as}(COO^-)$ and $\nu(COOH)$ bands respectively increase
or decrease, indicating further nitrate depletion due to more available $NO_3^-$ ions formation. In stage IV, the reaction

equilibrium is reached and $\Delta A$ values remain unchanged. In addition, corresponding IR spectra changes of 3:1 mixtures
during the shaded period are supplied in Fig. S8. At $\sim 69.2\%$ RH, the presence of 1741 $cm^{-1}$ band indicates the excess of
liquid OA, suggesting this displacement reaction tends to reach equilibrium with comparable final concentrations of
"reactants" and "products" (Wang and Laskin, 2014).

For the quantitative evaluation of phase state effect, the kinetics of this displacement reaction should be further

explored. For a given second-order reaction: $A + B \rightarrow P$ (P = products), the reaction rate can be determined as

$$\frac{d[A]}{dt} = -k[A][B] \tag{1}$$

where $k$ refers to the second-order rate constant ($cm^3$ $molecule^{-1}$ $s^{-1}$); [A] and [B] refer to the concentration of reactants A and
B, respectively. Herein, during stepwise RH increasing, the concentration of reactant A, i.e., aqueous $NO_3^-$, is limited and the
concentration of reactant B, i.e., liquid OA, is in excess, and hence, this displacement reaction can be treated as a pseudo

first-order reaction (He et al., 2017; Gao et al., 2018). The rate equation (1) can be rewritten as

$$\frac{d[A]}{dt} = -k_{app}[A] \tag{2}$$

And then,

$$[A] = [A_0]e^{-k_{app}t} \tag{3}$$

where $k_{app}$ is the pseudo first-order rate constant ($s^{-1}$) and equals to $k[B]$. $[A_0]$ refers to the initial concentration of reactant A.

Based on this, the concentrations of reactant A and products P will change exponentially. As well, the integrated absorbance
of 1620 $cm^{-1}$ ($\nu_{as}(COO^-)$) band can be used to describe the product concentration, thus the pseudo-first-order rate constant
$k_{app}$ can be determined by the exponential changes in absorbance difference of $\nu_{as}(COO^-)$ band, $\Delta \bar{A}$, as a function of reaction
time (Hung and Ariya, 2007; Hung et al., 2005; Gao et al., 2018). Namely, $\Delta \bar{A} = A_\infty e^{-k_{app}t}$, where $\Delta \bar{A} = A_t - A_\infty$, $A_t$ and
$A_\infty$ represent the integrated absorbance at time $t$ and time infinite, respectively.

As shown in Fig. 2, the $HNO_3$ release process can be divided into three stages, corresponding to three RH ranges, i.e.,
constant $\sim 15\%$, sudden RH increase from $\sim 14.8\%$ to $\sim 61.0\%$, and 66.2-69.2%. The measured $k_{app}$ values are shown in
Table 1 and the $\Delta \bar{A}$ changes with initialized reaction time are depicted in Fig. S9. As seen, the $HNO_3$ release is relatively
slow with a $k_{app}$ value of $\sim 1.70 \times 10^{-3}$ $s^{-1}$ at constant $\sim 15\%$ RH, possibly owing to limited $NO_3^-$ concentration and mass
transfer limitation within the particle phase; the humidification process at 14.8-61.0% RH exhibits the fastest $HNO_3$ release

with the $k_{app}$ value of $\sim 7.45 \times 10^{-2}$ $s^{-1}$; during $NaNO_3$ deliquescence, the $HNO_3$ release rate slightly decreases but is still
about one order of magnitude higher than that at constant $\sim 15\%$ RH. It is noteworthy that the $R^2$ is only 0.812 at 66.2-69.2%
RH, implying this displacement reaction may no longer obey to pseudo first-order reaction after $NaNO_3$ deliquescence due to
the formation of large amounts of available $NO_3^-$ ions, in other words, the concentration of aqueous $NO_3^-$ is no longer limited.
Furthermore, the fractions of reacting liquid OA at the end of three RH ranges are measured by the ratio of integrated

absorbance of 1741 $cm^{-1}$ ($\nu(COOH)$) band at a certain time to that before the reaction, which show a value of $\sim 32.3\%$ before



NaNO$_3$ deliquescence and reach ~ 71.0% after deliquescence. These results further confirm that aqueous NaNO$_3$ tends to be more reactive to liquid OA than amorphous and semisolid NaNO$_3$ due to the presence of large amounts of available NO$_3^-$ ions.

**Table 1** The pseudo-first-order rate constant $k_{app}$ and fractions of reacting liquid OA corresponding to different time periods and RH ranges.

| Time periods (s) | RH ranges (%) | $k_{app}$ (s$^{-1}$) | R$^2$ | Fractions of reacting liquid OA (%) |
|---|---|---|---|---|
| 4042-5731 | ~ 15 (constant) | $1.70 \times 10^{-3}$ | 0.991 | 6.9 |
| 7320-7374 | 14.8-61.0 | $7.45 \times 10^{-2}$ | 0.958 | 32.3 |
| 7570-7651 | 66.2-69.2 | $2.77 \times 10^{-2}$ | 0.812 | 71.0 |

A summary of phase state changes and chemical compositional evolution of NaNO$_3$/OA mixed particles is shown in Fig. 3. During the rapid drying (i.e., vacuuming process), mixed droplets release water and then OA dihydrate forms. After that, mixed particles are effloresced to produce amorphous NaNO$_3$ solids, and OA dihydrate is converted into anhydrous OA. At the same time, the aqueous NaHC$_2$O$_4$ is formed accompanied by the release of gaseous HNO$_3$. For the humidification process, first, amorphous NaNO$_3$ solids at < 5% RH are inert to liquid OA, due to the unavailable dissociative NO$_3^-$ ions for HNO$_3$ liberation. As RH increases to around 15%, amorphous solids are converted into viscous semisolids, which can exhibit chemical reactivity to liquid OA causing nitrate depletion. When the NaNO$_3$ is deliquesced, more available NO$_3^-$ ions are formed, and thus aqueous NaNO$_3$ is more reactive to liquid OA causing higher nitrate depletion extent. In previous studies, Li et al. (2017) indicated that the reaction between MA and NaCl could occur when aqueous H$^+$ from MA and Cl$^-$ ions from NaCl were available for HCl liberation, meanwhile, the reaction would slow down or stop once the amount of available H$^+$ became small or the particles were effloresced. Besides, Kuwata and Martin (2012) investigated the phase state effect of atmospheric SOM on its reactivity upon ammonia exposure. They found that the semisolid adipic acid and α-pinene SOM could take up small amounts of ammonia even at low RH. At high RH, the particles existed in liquid state, and the absorbed water could act as a plasticizer which decreased the viscosity of particles and increased the diffusion coefficient of ammonia, thereby leading to extensive ammonia uptake. It was noteworthy that the ammonia uptake was also thermodynamic or kinetically limited, showing that aqueous SOM particles were not fully neutralized even for the highest NH$_3$ concentration.





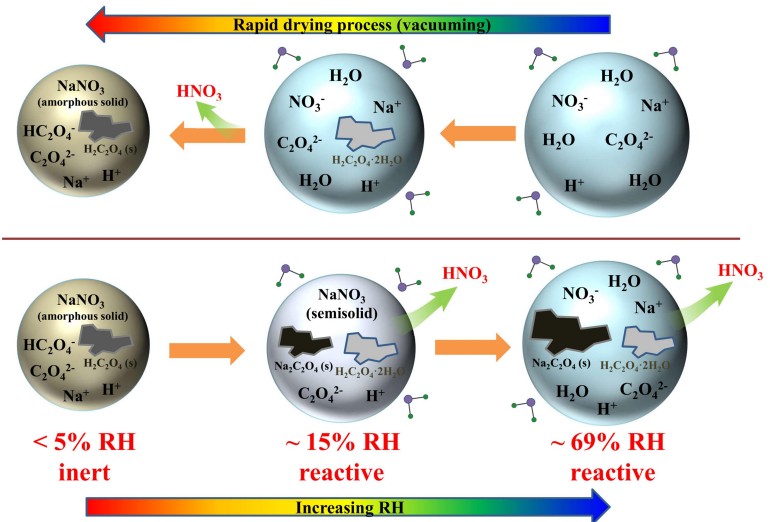

**Figure 3.** Schematic diagram of phase state changes and chemical compositional evolution of NaNO₃/OA mixed particles in the vacuuming and humidification processes.

**3.3 Hygroscopic growth and chemical composition evolution of NaNO₃/MA mixtures**

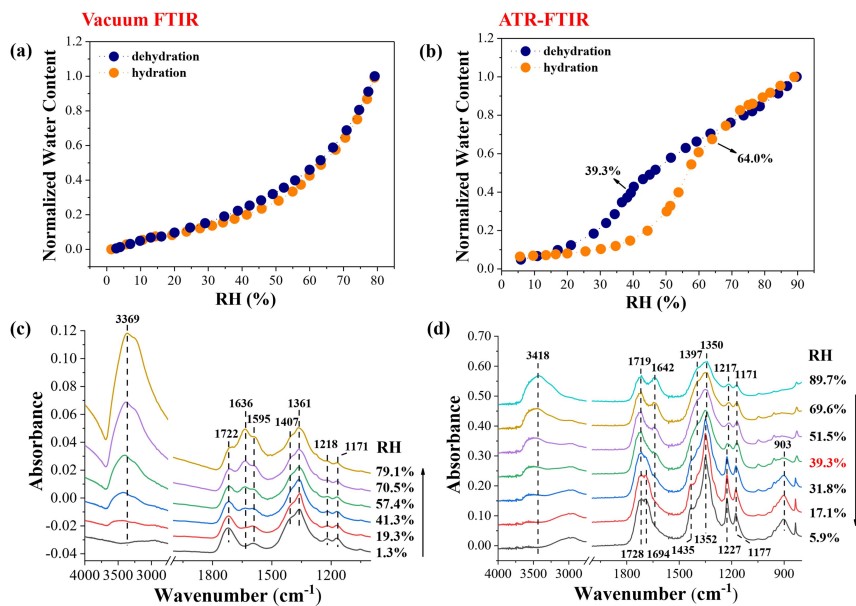

**Figure 4.** Hygroscopic growth curves of 1:1 mixed NaNO₃/MA particles measured by vacuum FTIR (a) and ATR-FTIR (b), as well as corresponding IR spectra during the humidification in vacuum FTIR measurement (c) and during the dehumidification in ATR-FTIR measurement (d).

Figure 4a and 4b display the hygroscopic behaviour of mixed NaNO₃/MA particles deposited on CaF₂ windows and Ge substrate measured by vacuum FTIR and ATR-FTIR, respectively. Correspondingly, the IR spectra upon hydration and upon dehydration are shown in Fig. 4c and 4d. Note that the IR spectra upon dehydration in vacuum FTIR measurement and upon hydration in ATR-FTIR measurement are shown in Fig. S10a and S10b, respectively. In vacuum FTIR measurement, the water content of mixed particles increases or decreases continuously with changing RH without distinct phase transitions (seen Fig. 4a). In other words, the addition of MA can totally inhibit the NaNO₃ crystallization. Likewise, Braban and Abbatt (2004) and Parsons et al. (2004) did not observe the efflorescence of (NH₄)₂SO₄ or MA in 1:1 mixed (NH₄)₂SO₄/MA particles under dry conditions. Also, no efflorescence was observed by Ghorai et al. (2014) for 1:1 NaCl/MA mixed system. In Fig. 4c, the 1722 cm⁻¹ band is assigned to $\nu$(C=O) of aqueous MA, and the 1407, 1218 and 1171 cm⁻¹ bands are also the characteristics of aqueous MA (Shao et al., 2017). The 1361 cm⁻¹ band is assigned to $\nu_3$(NO₃⁻) of NaNO₃ solution phase. More importantly, a new band located at 1595 cm⁻¹ indicates the formation of monosodium malonate (HOOCCH₂COONa)



(Wang et al., 2019; Shao et al., 2018). Thus, the displacement reaction between $NaNO_3$ and MA is confirmed by vacuum FTIR measurement as follows:

$$HOOCCH_2COOH(aq) + NaNO_3(aq) \rightarrow HOOCCH_2COONa(aq) + HNO_3(g)(\uparrow) \tag{R3}$$

The production of monosodium malonate has also been observed in mixed NaCl/MA aerosols by Li et al. (2017). They explained that the first acid dissociation constant ($pK_{a1}$) of MA was about 2.83, which was ~ 3 orders of magnitude larger than the second one ($pK_{a2} = 5.70$), resulting in much more $HOOCCH_2COO^-$ dissociated from MA than $CH_2(COO)_2^{2-}$ (Li et al., 2017). To further validate this displacement reaction, the IR spectra of $NaNO_3$/MA mixtures before and after the RH cycle are supplied in Fig. S11. It is clear that the 1722 cm$^{-1}$ band assigned to $\nu$(COOH) of MA becomes weaker, meanwhile,

the 1595 cm$^{-1}$ band assigned to $\nu_{as}$(COO$^-$) of monosodium malonate becomes stronger after the RH cycle.

     In ATR-FTIR measurement, deposited particles on the Ge substrate show distinct efflorescence and deliquescence transitions during the RH cycle. The initial efflorescence relative humidity (ERH) is ~ 39.9% and the RH where the particles are fully deliquesced is ~ 64.0%. As shown in Fig. 4d, the IR features of mixtures at 89.7% RH are comparable to that measured by vacuum FTIR. Whereas, as RH decreases to 39.3%, the 1719 cm$^{-1}$ band, assigned to $\nu$(COOH) of aqueous MA,

is split into two shoulder peaks located at 1728 and 1694 cm$^{-1}$. Furthermore, the 1397, 1217, and 1171 cm$^{-1}$ bands red-shift to 1435, 1227, and 1177 cm$^{-1}$, respectively. A new peak located at 903 cm$^{-1}$ appears. All these scenarios indicate the liquid-solid phase transition of MA (Shao et al., 2017). The 1350 cm$^{-1}$ band assigned to $\nu_3$(NO$_3^-$) of aqueous $NaNO_3$ turns into a sharp peak at 1352 cm$^{-1}$, suggesting the formation of $NaNO_3$ solids. In other words, the efflorescence of MA and $NaNO_3$ occurs synchronously at ~ 39.3% RH. As compared to vacuum FTIR results, we can infer that the heterogeneous efficacy of

Ge substrate is much higher than $CaF_2$ windows, resulting in heterogeneous nucleation of MA and $NaNO_3$ upon dehydration (Ma et al., 2021). As shown in Fig. 4b and S10b, the solid-liquid phase transition of crystalline MA and $NaNO_3$ ends at ~ 64.0% RH, which is lower than the deliquescence points of pure components. Indeed, the aerosol DRH can be significantly reduced by the mixing of organic acids and inorganic salts (Marcolli et al., 2004). Specially, no IR features of malonate sodium salts are observed, indicative of no nitrate depletion in ATR-FTIR measurement, which is differed from the vacuum

FTIR observation. The causes for this discrepancy will be discussed detailed in Sec. 3.5. Besides, the chlorine depletion in 1:1 mixed NaCl/MA particles with two different RH changing rates is also experimentally detected to further probe the influence factors for acid-displacement reactions, and the results are discussed in the Supplement.



**3.4 Hygroscopic growth and chemical composition evolution of NaNO₃/GA mixtures**

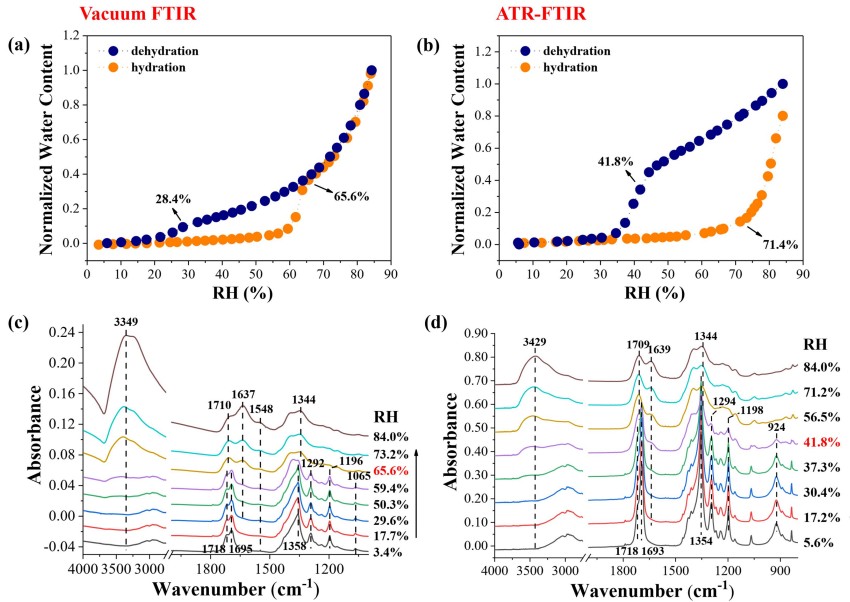

**Figure 5.** Hygroscopic growth curves of 1:1 mixed NaNO₃/GA particles measured by vacuum FTIR (a) and ATR-FTIR (b), as well as corresponding IR spectra during the humidification in vacuum FTIR measurement (c) and during the dehumidification in ATR-FTIR measurement (d).

The hygroscopic growth curve and corresponding IR spectra of 1:1 mixed NaNO₃/GA particles measured by vacuum FTIR are shown in Fig. 5a and 5c, respectively. Routinely, mixed particles are first dried in vacuum and then undergo a RH cycle.

In Fig. 5c, at ~ 3.4% RH, the shoulder peaks located at 1718 and 1695 cm⁻¹ are attributed to $v$(C=O) of crystalline GA, and the $v$(C-O) band at 1292 cm⁻¹ and rocking vibration mode of CH₂ ($\gamma$(CH₂)) located at 1196 cm⁻¹ are also the characteristics of GA solids (Wu et al., 2019a). The 1358 cm⁻¹ band is attributed to $v_3$(NO₃⁻) of NaNO₃ solids. Indeed, mixed NaNO₃/GA particles exhibit distinctly different efflorescence behaviour compared with NaNO₃/MA mixtures, considerably consistent with the observation by Ghorai et al. (2014) for NaCl/GA(MA) mixed systems. This may be attributed to weaker inhibiting effect of GA than MA on inorganic salt crystallization (Ma et al., 2021). In addition, a weak peak located at 1548 cm⁻¹ appears upon hydration, indicating the formation of glutarate sodium salts. The weaker intensity compared with NaNO₃/MA system indicates the substantially weaker chemical reactivity of GA ($pK_{a1}$ = 4.32) than MA ($pK_{a1}$ = 2.83) (Ghorai et al., 2014). As RH increases to 65.6%, mixed NaNO₃/GA particles are fully deliquesced, and corresponding IR feature changes are consistent with the observation by Wu et al. (2019b). During the dehumidification, aqueous droplets are effloresced at ~



28.4% RH, judged mainly from the IR feature changes of mixtures shown in Fig. S13a. It should be noted that only $v_3(NO_3^-)$
band experiences a red shift from 1344 to 1358 cm$^{-1}$, implying only NaNO$_3$ component effloresces and GA cannot be
crystallized upon dehydration. The particle morphology of partial crystallization for mixtures of GA and nitrates was also
observed by Wang and Laskin (2014). Braban (2004) found that GA would not effloresce upon drying for (NH$_4$)$_2$SO$_4$/GA
mixed system, potentially owing to the residual (NH$_4$)$_2$SO$_4$ in solution phase added extra barrier to the formation of GA

crystalline germ (Braban and Abbatt, 2004). Wu et al. (2019b) studied the stepwise efflorescence process of 1:1 mixed
NaNO$_3$/GA particles in the pulsed RH mode with vacuum FTIR method. They found that effloresced particles reversibly
absorb and release water with pulsed RH changes when the minimal RH values were below 10%, suggesting there were still
small amount of water retained in mixed particles, i.e., the mixed particles were partially crystallized. In addition, the
crystallization of GA in the vacuuming process can be explained by the lower temperature of droplets arises from rapid

water evaporation, as discussed in the Supplement.

      In ATR-FTIR measurement, the efflorescence of mixed NaNO$_3$/GA particles occurs at ~ 41.8% RH, showing an abrupt
decrease of water content with decreasing RH (seen Fig. 5b). For the humidification process, mixed particles obviously take
up water at ~ 71.4% RH. The water content cannot match with that in the dehumidification process even until ~ 84.0% RH,
suggesting the GA component cannot be deliquesced completely due to its high DRH (Marcolli et al., 2004; Yeung et al.,

2010). This is further confirmed by corresponding IR features shown in Fig. S13b. As seen, the feature bands at 1207 and
924 cm$^{-1}$ assigned to GA crystals are still observed at RH as high as 84.0%. Likewise, partial dissolution of GA was also
observed for the (NH$_4$)$_2$SO$_4$/GA system by Ling and Chan (2008). More importantly, there are no IR features of glutarate
sodium salts observed during the RH cycle, which will be further explained in Sec. 3.5.

**3.5 Influence factors for acid-displacement reactions**

As already indicated, the driving out of gaseous HCl/HNO$_3$ from chloride/nitrate and organic acids mixtures is mainly
dominated by two factors, i.e., $K_{a1}$ and $K_H$ of organic acids (Wang and Laskin, 2014; Laskin et al., 2012). Note that the acid
dissociation constant ($pK_{a1}$, $pK_{a2}$), equilibrium constant $K_1$, $K_2$ and $K_3$ utilized in the pH change simulation, as well as
Henry's law constant ($K_H$) at 298 K for different acids involved in this study are listed in Table 2. In our previous review, the
chlorine depletion in mixed NaCl/diacids systems was suggested to be related to the acidity and environmental concentration

of organic acids, as well as ambient RH and particle size of mixtures (Chen et al., 2021). First, stronger acidity would cause
more available H$^+$ in aqueous phase, favouring the acid-displacement reaction. The acidity of diacids followed the order of
OA > MA > SA > GA. Then, the higher acid concentration and lower RH would lead to greater HCl partial pressure, which
was favourable for the partitioning of HCl into the gas phase. Finally, the chlorine depletion extent, $\xi$, was determined to be
inversely proportional to $r^2$ where $r$ was the droplet radius. In other words, the depletion extent $\xi$ would increase greatly with

decreasing droplet size.



As known, the driving out of volatile species such as $HNO_3$ from aqueous droplets can be quantified by Maxwell steady-state diffusive mass transfer equation (Chen et al., 2021; Cai et al., 2014; Ray et al., 1979). The mass flux of $HNO_3$ partitioning from particle to gas phase can be determined as

$$-\frac{dm}{dt} = \frac{4\pi rMD}{RT}(p_\infty - p_r) \tag{4}$$

where $m$ denotes the mass of $HNO_3$ within the droplets (g); $t$ denotes the evaporation time (s); $M$ and $D$ represent the molecular weight (g mol$^{-1}$) and diffusion coefficient of $HNO_3$ in the air (m$^2$ s$^{-1}$), respectively; $R$ is the ideal gas constant (J mol$^{-1}$ K$^{-1}$); $T$ is the temperature (K); $r$ is the droplet radius (m); $p_\infty$ and $p_r$ represent the partial pressure of $HNO_3$ (Pa) at infinite distance and droplet surface, respectively.

When the RH decreases continuously, the evaporation rate of $HNO_3$ at a certain RH, $k_{RH}$ (g s$^{-1}$), can be expressed as

(assuming $p_\infty = 0$)

$$k_{RH} = \frac{dm}{dt} = \frac{4\pi rMD}{RT}p_r \tag{5}$$

In the ATR-FTIR measurement, the diffusion coefficient of $HNO_3$ in the air is $(1.18 \pm 0.03) \times 10^{-5}$ m$^2$ s$^{-1}$ under the condition of $T = 298$ K and $P = 1$ atm (Durham and Stockburger, 1986). Assuming that the droplet radius at 90% RH is ~ 1.5 μm, the droplet size $r$ at any RH can be determined according to the size growth factors predicted by the Extended Aerosol

Inorganic Model (E-AIM). $p_r$ can also be estimated by the E-AIM model. Note that the E-AIM predictions are performed by UNIFAC model with parameters modified by Peng et al. (2001). In addition, the RH can be converted to water activity, $a_w$, by Köhler equation to minimize the Kelvin curvature effect, as shown in the Supplement (Jing et al., 2016). Based on these, the $k_{RH}$ for 1:1 mixed $NaNO_3$/OA, $NaNO_3$/MA, and $NaNO_3$/GA systems as a function of $a_w$ in ATR-FTIR measurement can be estimated and shown in Fig. 6a. Note that the solid phase formation is prevented to obtain the simulation data of

supersaturated droplets at low RH.

Besides, according to Chapman-Enskog method, the diffusion coefficient of $HNO_3$ in the gas phase can be expressed as (Reid et al., 1987)

$$D = \frac{0.00266T^{3/2}}{PM_{AB}^{1/2}\sigma_{AB}^2\Omega_D} \tag{6}$$

where subscripts A and B denote species A and B; $P$ is the ambient pressure (bar); $M_{AB} = 2[(1/M_A) + (1/M_B)]^{-1}$ (g mol$^{-1}$); $\sigma_{AB}$

represents the characteristic length (Å); $\Omega_D$ is the diffusion collision integral. Thus, the $HNO_3$ diffusion coefficient in vacuum FTIR measurement, $D^*$, is much higher than that in ATR-FTIR measurement, due to the much lower ambient pressure in vacuum FTIR, $P^*$, which can be expressed as

$$P^* = P_0 * RH \tag{7}$$

where $P_0$ denotes the saturated water vapor pressure at 298 K (bar). Based on equations (6) and (7), the $D^*$ value at any RH

can be determined. Accordingly, the $k_{RH}$ values in vacuum FTIR measurement can also be calculated for comparison, as shown in Fig. 6a. The $k_{RH}$ values in vacuum FTIR measurement are ~ 35 times higher than those in ATR-FTIR measurement at $a_w = 0.9$, due to much lower ambient pressure in vacuum FTIR. Based on this, the discrepancies among the two





measurements for nitrate depletion in NaNO$_3$/MA(GA) systems can be mainly attributed to the higher HNO$_3$ release rate arises from faster HNO$_3$ gas phase diffusion in vacuum FTIR. Previous studies have indicated that numerous atmospheric

processes involved aerosol particles were often carried out at high altitudes with significantly lower pressure than the ground level (Zhao et al., 2009; Rosenberger et al., 2018; Schilling and Winterer, 2014), and aerosol properties such as hygroscopicity under reduced pressure should be further characterized (Tang et al., 2019). Therefore, the ambient pressure, dominating the diffusion coefficient of HNO$_3$ in the gas phase, tends to play an important role in nitrate depletion during the transport and aging of atmospheric aerosols. Furthermore, mixed NaNO$_3$/OA system exhibits higher HNO$_3$ release rate than

NaNO$_3$/MA(GA) systems due to the higher acidity of OA. Meanwhile, the HNO$_3$ release rate for NaNO$_3$/OA system in ATR-FTIR measurement is comparable to that for NaNO$_3$/GA system in vacuum FTIR measurement. These indicate that the lower acidity and hence lower reactivity of MA and GA also contribute to the unobserved nitrate depletion in ATR-FTIR measurement. In addition, all the $k_{RH}$ values increase significantly with decreasing droplet water activity. Specifically, the $k_{RH}$ at $a_w = 0.1$ is about two orders of magnitude higher than that at $a_w = 0.9$. It should be noted that in ATR-FTIR

measurement, HNO$_3$ release from NaNO$_3$/MA(GA) aqueous droplets at very low RH would exhibit comparable levels to that for NaNO$_3$/OA system at relatively higher RH. Therefore, considerable HNO$_3$ release from NaNO$_3$/MA(GA) mixtures can be expected in case of no complete efflorescence of mixtures in ATR-FTIR measurement. In other words, the crystallization of mixed droplets induced by the Ge substrate, causing the lack of aqueous H$^+$ and NO$_3^-$ ions available for HNO$_3$ liberation, tends to be another cause for negligible or even no HNO$_3$ release from NaNO$_3$/MA(GA) mixtures.

Besides, the pH changes of 1:1 NaNO$_3$/DCAs aqueous droplets with initial concentration of 0.1 mol/L as a function of nitrate depletion fraction are calculated and shown in Fig. 6b. The calculation method is similar to that for ammonium depletion simulation proposed by Wang et al. (2019) (shown in the Supplement). As the NO$_3^-$ in aqueous phase is depleted, the conversion of DCAs to their sodium salts proceeds, leading to the continuous reduction in droplet acidity.

**Table 2** The acid dissociation constant ($pK_{a1}$, $pK_{a2}$), equilibrium constant $K_1$, $K_2$ and $K_3$ utilized in the pH change simulation, as well as
Henry's law constant ($K_H$) at 298 K for different acids involved in this study.

| Species | $pK_{a1}$[a] | $pK_{a2}$[a] | $K_1$ ($K_{a1}$) | $K_2$ ($K_{a2}$) | $K_3$ (1/$K_{a1}$) | $K_H$ (M/atm)[b] |
|---|---|---|---|---|---|---|
| HNO$_3$ | -1.27 | | $> 2 \times 10^1$ | | $5 \times 10^{-2}$ | $> 2 \times 10^5$ |
| Oxalic acid | 1.23 | 4.19 | $5.9 \times 10^{-2}$ | $6.5 \times 10^{-5}$ | | $(0.06\text{–}6.2) \times 10^8$ |
| Malonic acid | 2.83 | 5.70 | $1.5 \times 10^{-3}$ | $2.0 \times 10^{-6}$ | | $(0.26\text{–}3.3) \times 10^{10}$ |
| Glutaric acid | 4.32 | 5.42 | $4.8 \times 10^{-5}$ | $3.8 \times 10^{-6}$ | | $(0.35\text{–}3.3) \times 10^9$ |

[a]: Data from Haynes and Lide (2011).

[b]: Data from Wang and Laskin (2014), Compernolle and Müller (2014), Soonsin et al. (2010), and Bilde et al. (2003).





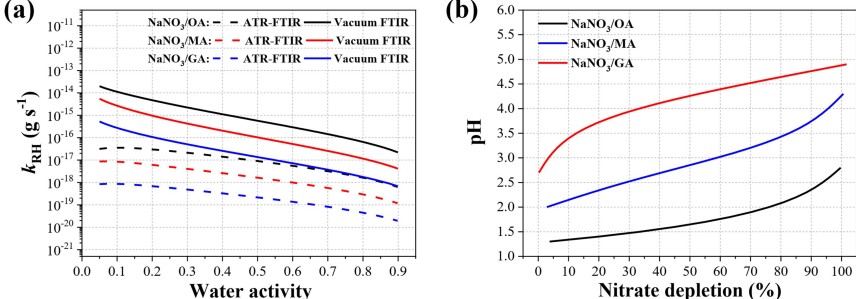

**Figure 6.** (a) The $k_{RH}$ values for 1:1 mixed droplets of NaNO$_3$ and DCAs as a function of water activity at 298 K in the two measurements, i.e., ATR-FTIR (dashed lines) and vacuum FTIR (solid lines). Note that the droplet radius is assumed to be ~ 1.5 μm at $a_w$ = 0.9. (b) The droplet pH of 1:1 NaNO$_3$/DCAs mixtures with initial concentration of 0.1 mol/L as a function of nitrate depletion fraction at 298 K.

## 4 Conclusions and atmospheric implications

The conversion of organic acids to organic acid salts and the internal heterogeneity can significantly alter the hygroscopic properties, acidity, optical properties, viscosity, and chemical reactivity of mixed aerosols (Ma et al., 2013, 2019b; Peng et al., 2016; Ghorai et al., 2014). However, the formation and precipitation of organic acid salts have not been considered by the current version of E-AIM. In this work, the hygroscopic behaviour and chemical composition modification of NaNO$_3$/DCAs mixtures during the RH cycle were investigated by vacuum FTIR and ATR-FTIR techniques. First, OA could react with nitrate causing considerable nitrate depletion in both the measurements. Indeed, substantial metal oxalate complexes, which could not contribute to the CCN activity of aerosols, were frequently detected mixed in atmospheric aerosols (Furukawa and Takahashi, 2011). At the same time, the nitrate phase state would play a critical role in determining the occurrence and extent of nitrate depletion, e.g., amorphous NaNO$_3$ solids were inert to liquid OA; at around 15% RH, some moisture was absorbed to form semisolid NaNO$_3$ accompanied by HNO$_3$ liberation; the nitrate depletion extent and HNO$_3$ release rate increased with increasing RH; during NaNO$_3$ deliquescence, the displacement reaction proceeded due to more available NO$_3^-$ ions formation. In addition, NaNO$_3$ can be treated as a surrogate for a broad class of amorphous or semisolid species existing in atmospheric aerosols, which undergo homogeneous or heterogeneous reactions causing secondary inorganic or organic aerosols formation. Therefore, the phase state effect may also be suitable for relevant aerosol reaction systems.

MA and GA exhibited weaker chemical reactivity to nitrate than OA in vacuum FTIR measurement. Indeed, in supermicrometer sea-salt aerosols, the chloride depletion was mainly attributed to sulfate and nitrate, followed by methanesulfonate and oxalate, while malonate and glutarate played a minor role in Cl$^-$ loss (Kerminen et al., 1998). More importantly, almost no sodium salts of MA and GA were produced in ATR-FTIR measurements, differing from the vacuum



FTIR observations. This discrepancy was confirmed to be mainly dominated by the faster $HNO_3$ gas phase diffusion arises from much lower ambient pressure in vacuum FTIR. Besides, the weaker acidity of MA and GA, lower $HNO_3$ release rate at higher RH, as well as the occurrence of efflorescence transition of aqueous droplets also contributed to the unobserved

nitrate depletion in ATR-FTIR measurements. These scenarios indicate that organic acids have a potential to deplete nitrate based on the comprehensive consideration of acidity, particle-phase state, droplet water activity, as well as $HNO_3$ gas phase diffusion (influenced by ambient pressure and so on). Our results reveal that faster $HNO_3$ gas phase diffusion, higher acidity of organic acids, lower droplet water activity, as well as the absence of efflorescence transition upon drying would be favourable for $HNO_3$ partition into the gas phase.

Besides, this work will help us understand the discrepancies among previous observations on chloride/nitrate depletion. For instance, in Ghorai's study, the submicrometer NaCl/MA(GA) particles deposited on $Si_3N_4$ windows and TEM grids tended to stay metastable or liquid-like state under dry conditions, leading to no or partial crystallization of mixtures which facilitated the HCl liberation (Ghorai et al., 2014). Likewise, the micron-sized $NaNO_3$/MA(GA) particles deposited on carbon-filmed grids and $Si_3N_4$ windows would exist in amorphous liquid or viscous semisolid state after drying, causing

significant $HNO_3$ liberation (Wang and Laskin, 2014). In contrast, Ma et al. (2013) did not observe the chlorine depletion in mixed NaCl/MA(GA) particles placed in aluminum sample holder and deposited on Ge substrate, which might be attributed to the heterogeneous nucleation of mixed droplets in the fast drying process. Also, the formation of $Na_2C_2O_4$ was observed in internally mixed NaCl/OA particles due to the higher acidity of OA (Ma et al., 2013).

*Data availability.* All data are available upon request from the corresponding author.

*Author contributions.* SSM and YHZ designed the experimental plan. SSM and QL preformed the measurements. QL helped with data analysis. SSM and YHZ wrote the paper. All authors discussed and contributed to the manuscript.

*Competing interests.* The authors declare that they have no conflict of interest.

*Acknowledgements.* This work was supported by the National Natural Science Foundation of China (Nos. 42127806 and 41875144).

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
