# Peer review of "A comprehensive study on hygroscopic behaviour and nitrate depletion of NaNO3 and dicarboxylic acid mixtures: Implications for nitrate depletion in tropospheric aerosols"

_Atmospheric Chemistry and Physics, 2022_

## Referee Comment (RC2)

In this work by Shuaishuai Ma, Qiong Li and Yunhong Zhang, the water uptake and nitrate/nitric acid depletion of laboratory particles are studied experimentally and evaporation rates estimated with a kinetic model. The authors employed ATR-FTIR and vacuum FTIR methods to study their particles deposited on substrates exposed to different relative humidities (RH). The main focus is on the spectroscopic signatures of nitrate species, nitrate salt formation, and other phase changes when 1:1 mixtures of $NaNO_3$ and dicarboxylic acids (DCAs) undergo RH cycles.

Overall, this work is very interesting and within the subject area of this journal. The manuscript is also well structured and for the most part well written (a few English phrasing issues will need to be addressed).

I have one major comment and many minor ones (see below) that the authors should address to clarify which processes are backed by direct evidence and which ones are based on assumptions or inferred rather indirectly.

**Major comments**

1. My main concern is with the presentation and interpretation of several of the results with regard to the loss of nitrate from the particles/droplets via evaporation of $HNO_3$. As the title indicates, nitrate depletion is a key interest of this work, yet most of the data provided does not appear to show direct evidence for nitrate depletion via $HNO_3$ loss to the gas phase – contrary to what is stated in the abstract (line 15) and the Results and Discussion section.

   The measurements provided characterize the composition and spectral features of nitrate and DCA species in the condensed phase(s) of deposited particles. However, unless I missed it, $HNO_3(g)$ is not measured, nor is the mass of nitrate in the particle phase tracked. If there is substantial loss of nitrate, the particles should change in nitrate content and overall mass. Moreover, the loss of nitrate would likely be accompanied by a loss of water. As pointed out in the specific comments that follow, there are several instances where statements like "... confirms nitrate depletion and $HNO_3$ release" are made (e.g. line 133), while the information provided does not directly support this conclusion. For example, the authors seem to equate the formation of liquid or solid $NaHC_2O_4$ as evidence for nitrate depletion. However, it is not discussed why that should be a unique signature of such a process. This reviewer argues that (at least some) dissolved or crystalline $NaHC_2O_4$ could also form in the absence of any nitrate depletion, simply as a result of aqueous solution chemistry and resulting equilibria. I strongly suggest that the authors rectify this issue and provide direct evidence for $HNO_3$ loss to the gas phase or otherwise clarify that their interpretation depends on certain assumptions about the relevant processes taking place.

   To be clear, I also consider nitrate depletion via $HNO_3$ to be a plausible hypothesis and a likely explanation for the findings from this work. However, what is missing is quantitative evidence and/or clarity in the discussion in support of this hypothesis.

**Specific comments**

2. Title: phrasing of "Implication for the influence factors of nitrate depletion" needs language improvement and perhaps a more direct link to aerosols. Consider replacing by "implications for nitrate depletion in tropospheric aerosols".

3. Line 15: make statement consistent with the data provided (see my major comment).

4. Line 19: "... potentially indicating the transformation of amorphous solids to semisolid NaNO3; ..."; consider: or the dissolution of $NaNO_3$ into a liquid DCA + Water rich phase?

5. Line 39: define abbreviations like DCA, ATR-FTIR and OA at first use in the main text (aside from the abstract).

6. Line 42: there are several directly relevant studies on hygroscopicity and surface tension effects that could be cited here; e.g. Peng et al. (2001, 10.1021/es0107531), Ovadnevaite et al. (2017, doi:10.1038/nature22806), Hodas et al. (2015, doi:10.5194/acp-15-5027-2015).

7. Line 43: "It is well known that the displacement of strong acids, i.e., HCl or HNO3, by weak organic acids, e.g., DCAs, is not thermodynamically favoured in bulk solution". This statement requires citation of a relevant reference. Note that even if the thermodynamic equilibrium is favouring the left hand side of reaction (R1), there will be some amounts of HCl or $HNO_3$ dissolved and if gas phase exchange is possible, there will be a continuous loss of the right hand side, consuming the reactants of R1 over possibly long time (depending on bulk vs. small aerosols). Therefore, given that in an aqueous phase, $NaNO_3$ will be present mostly in the form of $Na^+$ and $NO_3^-$, some NaA(aq) (or solid) and/or $Na^+$ and $A^-$ may be present even in the absence of any evaporation of $HNO_3$.

8. Line 52: State for what concentration/conditions. Are you comparing at the same RH or the same water content or the same molar concentration of dissolved solutes?

9. Line 58: define the base of the logarithm and perhaps at first use, define the meaning of $pK_{a1}$ formally using molal activities or standard chemical potentials, as appropriate given the values stated.

10. Line 66: what "substrate" is here referred to? do you mean material onto which particles were deposited in laboratory experiments?

11. Line 70: "... the substrate supporting OA droplets would crystallize to form"; do you mean the droplet / OA or really the substrate? Clarify phrasing.

12. Line 70: "OA dihydrate at 71% RH" Is it under conditions of dehumidification? Describe what kind of laboratory experiment you refer to.

13. Line 72: "Pure NaNO3 droplets might not be effloresced..." By "be effloresced" do you rather mean "crystallize" here? Note that effloresced is not necessarily the same as crystalline in the context of amorphous solids as a possibility. Also, clarify under what conditions; here presumably dehydration in an isothermal (and isobaric) experiment.

14. Line 73: "certain RH values" Which ones? Please provide values/ranges.

15. Line 74: Given the stated list of options, how is "highly viscous" different from glassy or semisolid? One could say "highly viscous (i.e. semisolid or glassy), liquid, or mixed-phase (e.g. solid–liquid) mixing states.

16. Line 75: Koop et al. (2011, doi:10.1039/C1CP22617G) would be another key reference to cite here.

17. Line 82: phrasing: "would" or rather "does" it?

18. Line 85: delete "mixture"; it is redundant with solution.

19. Line 92: "... a brief was introduced here"; phrasing.

20. Line 103: "chamber, which could be used to calibrate the ambient RH"; in the context of this description, it is mentioned that water vapor and $CO_2$ were removed by the vacuum pump. So, how was the RH established in the sample chamber? Is it due to remaining water vapor under low pressure conditions? If so, does that mean that the sample droplets are (potentially) evaporating water alongside with $HNO_3$ and DCA? The droplets may also outgas dissolved $CO_2$ during the experiment given the vacuum applied, which might have some effect on the acidity.

21. Line 127: "and anhydrous OA" I suggest to mention that you refer to crystalline anhydrous OA here.

22. Line 131: "... indicative of the $NaHC_2O_4$..."; Is this referring to an actual solid $NaHC_2O_4$ component forming or rather the presence of partially dissociated OA anions ($HC_2O_4^-$) in aqueous solution alongside $Na^+$? In the latter case, this should not be referred to as $NaHC_2O_4$ since it is not forming a compound of that structure nor stoichiometry. Also, in that case, this does not confirm directly the release of $HNO_3$, it only indicated the presence of partially dissociated OA. Please discuss.

23. Line 132: "Likewise, Wang et al. (2017) observed the formation of $NH_4HC_2O_4$ in mixed $(NH_4)_2SO_4$/OA droplets upon drying. These scenarios confirm the nitrate depletion and $HNO_3$ release from $NaNO_3$/OA mixtures in the vacuuming process."
The provided data and discussion does not uniquely support this conclusion. While $HNO_3$ evaporation is possible, where is the evidence for it? Oxalate salts could form also without any loss of $HNO_3$ to the gas phase. For example, $HNO_3$(aq) could form or, more likely, $H^+$(aq) and $NO_3^-$(aq) alongside solid or dissolved OA salts. Please provide evidence or a more nuanced discussion.

24. Line 145: "Likewise, Ma et al. (2013) found that the DRH of NaCl component decreased in both external and internal NaCl/MA mixtures." How could this be the case in an external mixture (assuming external means (pure) NaCl particles separate from MA particles)?
Also, worth mentioning in the context of DRH lowering: a lowering of the DRH in particles containing aqueous DCAs is expected from thermodynamic equilibrium theory and has been shown in many other measurements and by means of thermodynamic model predictions, such as those shown for MA or OA and ammonium sulfate by Bouzidi et al. (2020, doi:10.1016/j.atmosenv.2020.117481) and those shown by Hodas et al. (2015, doi:10.5194/acp-15-5027-2015) for mixtures of ammonium sulfate with DCAs and citric acid.

25. Line 146: "Note that the OA dihydrate and crystalline Na2C2O4 cannot be deliquesced due to their very high DRHs ...".
A note: the process of deliquescence, as more clearly observed in binary (water + 1 solute) systems, should not be confused with partial (gradual) dissolution in multicomponent, multiphase systems. Therefore, please clarify the sentence. At present it is misleading. While crystalline substances like OA and Na2C2O4 may not fully dissolve, it should be noted that in the presence of an aqueous phase a solid–liquid equilibrium will be established (given enough time) and some amount of OA and Na2C2O4 will be in the dissolved state, forming a saturated aqueous solution with respect to the pertaining crystalline phase or phases; see, e.g. the discussion by Hodas et al. (2016, doi:10.5194/acp-16-12767-2016).

26. Line 155: "Upon hydration, the water content at high RH is far below that upon dehydration, ..."
This statement needs further clarification. It seems only to make sense when the dehydration refers to dehydration from an initial state of completely dissolved particles at very high RH. Of course, one could start dehydration at any RH and then the water content could be higher or lower than when arrived at that state from an initial state of low RH.

27. Line 159: I find Fig. S6 insightful and suggest that this figure and the related discussion be moved to the main manuscript.

28. Line 160: "Besides, as shown in Fig. 1c, the absorbance of 1620 $cm^{-1}$ band assigned to oxalate shows a slight increase at RH as low as 21.2%, implying the nitrate depletion proceeds at relatively low RH."
In my opinion, what is missing are observations of the loss of nitrate/$HNO_3$ from the particles. Do the authors have comparison examples for cases where the gas phase exchange is limited (small gas volume) such that only a small loss could occur and, as a result, the signature in oxalate formation substantially different? Without such observations, or other quantitative measures of loss to the gas phase, the interpretation of the experiments hinges on assuming a nitrate depletion takes place without having the data to confirm it. Please discuss.

29. Line 173: "As RH increases to $\sim$ 15%, the $\Delta A$ of $\nu_{as}$(COO-) band exhibits a considerable increase, indicative of the occurrence of nitrate depletion."
While indicative as an option, it remains speculative. It is indicative of the partial dissociation of OA in solution. Which is expected to increase with increasing particle water content and, hence, increasing RH.

30. Line 177: "2013). Then, the 1620 $cm^{-1}$ band appears and becomes stronger with time, suggesting $Na_2C_2O_4$ can be continuously produced at constant $\sim$ 15% RH."
I think this is better evidence for $Na_2C_2O_4$ formation and associated nitrate depletion, because the system is observed at constant RH, which means it should maintain constant water activity (in sufficiently large particles where the Kelvin effect would not change with evaporation). It may be good to expand the discussion of this piece of evidence.

31. Line 179: "... we can infer the conversion of amorphous $NaNO_3$ solids to highly viscous semisolids due to the uptake of trace amounts of moisture."
Please clarify: semisolids of what? if $NaNO_3$ dissolves partially and forms a viscous aqueous solution, this should not be called a $NaNO_3$ semisolid because the formed phase contains other species too.

32. Line 216: Define the $R^2$ metric.

33. Line 231: "As RH increases to around 15%, amorphous solids are converted into viscous semisolids, ...". Is there any evidence from the spectra for such a conversion and the approximate viscosity of the phase to classify it as semisolid?

34. Line 261: "... formation of monosodium malonate ...". Is this for a solid malonate phase or dissolved?

35. Line 266: Clarify: "that the first acid dissociation constant (pKa1) of MA was about 2.83, which was $\sim 3$ orders of magnitude larger than the second one ...". 3 orders of magnitude in what? clearly not in pKa. Also, 2.83 is the pKa1 not Ka1 (which the current phrasing implies).

36. Line 270: band assigned to $\nu_{as}(COO^-)$ of monosodium malonate. Please clarify: why of monosodium malonate and not of the malonate anion? The $COO^-$ functionality does not contain any sodium and in aqueous solution the malonate ion is likely dissociated from $Na^+$. Is the assigned band specific to the COO- group in crystalline monosodium malonate only?

37. Line 277: "aqueous $NaNO_3$" should be "aqueous $NO_3$.

38. Line 279: "As compared to vacuum FTIR results, we can infer that the heterogeneous efficacy of Ge substrate is much higher than CaF2 windows,...". It seems unclear whether this is really a substrate effect or rather an effect of the vacuum cell procedures applied.

39. Line 290, Fig. 5: What explains the difference in the normalized water contents comparing panels (a) and (b)? Did you compare the obtained water contents to model predictions of the water content or independent mass growth factor measurements to assess which technique provides more accurate hygroscopic cycles?

40. Line 301: "appears upon hydration, indicating the formation of glutarate sodium salts." Could you clarify whether this means that a solid crystalline salt phase is forming upon hydration in equilibrium with an aqueous solution?

41. Line 314: "lower temperature of droplets" Is it just the temperature effect or also the drying to much lower effective RH than during the ATR-FTIR measurement cycles?

42. Line 335: Could you discuss in this context whether the size dependence effect is due to the Kelvin effect, leading to a lower water activity and higher pH in smaller particles compared to larger ones exposed to the same RH in the gas phase, or some other effect?

43. Line 346, Eq. (5): check equation; it looks like the $p$ of $p_r$ is missing.

44. Line 358: Eq. (6), for consistency with Eq. (7), consider replacing $P$ by $p^*$.

45. Line 373: "Therefore, the ambient pressure, dominating the diffusion coefficient of HNO3 in the gas phase, tends to play an important role in nitrate depletion during the transport and aging of atmospheric aerosols."

This statement is not supported by any quantitative data. How big of a difference is expected in the troposphere? Obviously, one should not directly compare the vaccum FTIR pressure to realistic tropospheric conditions without accounting for the differences in the ranges of total pressures. If nitrate-containing particles are given hours to days to undergo displacement reactions, the impact of pressure on HNO3 diffusion may be unimportant. Do you have predictions to the contrary?

46. Line 409: "NaNO3 can be treated as a surrogate for a broad class of amorphous or semisolid species existing in atmospheric aerosols,..."
Please state in what RH range this semisolid state applies. NaNO3 is certainly dissolved and liquid-like in dilute aqueous aerosols at elevated RH (for aqueous NaNO3 viscosity data see e.g. Baldelli et al., 2016, doi:10.1080/02786826.2016.1177163; Lilek and Zuend, 2022, doi:10.5194/acp-22-3203-2022).

47. Line 427: "metastable or liquid-like state..." Please rephrase; metastable and liquid-like characterize two different properties. One related to the supersaturation of aqueous solutions with respect to a certain crystalline phase, while the other refers to a phase state (viscosity related).

**Supplement: minor comments**

47. SI, lines 20–36: How was that ERH determined? Was it based on just some features of the spectra? Fig. S1a does not seem to indicate any clear signature of efflorescence. Specifically, note that in the absence of severe kinetic limitations, one would expect a sharp change in mass growth factor at the ERH (point of crystallization) of a single-solute particle. Certainly so for a system at equilibrium with the gas phase. Why is that not the case for the systems shown in Fig. S1? Figure S2 shows also no indication of NaNO3 crystallization. Do the authors use the term "efflorescence" for a phase transition other than crystallization? If so, that would be untypical given the common use and meaning of this term and might need clarification in the text.

48. SI, line 40: what about that of a NaNO3 hydrate crystal? Would equilibrium thermodynamics not require that amorphous solids take up some water upon hydration (while single-solute crystalline particles may not)?

49. SI, lines 155–158, 166: these sentences are inconsistent. One is about HCl release, the other about HNO3 but referring to the same study. Please clarify.

50. SI, line 178: molecular weight (or rather molar mass) of water should be stated in $kg\,mol^{-1}$ to stay consistent with the other units given. Other there will be an incorrect Kelvin effect value.

51. SI, line 188: It should be stated that the true equilibria in aqueous solution involve use of the activities of the species, not the molar concentrations. Also, on line 210, state that self-dissociation of $H_2O$ is not considered.

52. SI, line 216: given definition of pH; note that this refers only to an approximate, molarity-based pH, as discussed by Pye et al. (2020, doi:10.5194/acp-20-4809-2020). The proper definition of pH involves the molality-based activity of $H^+$.

---

## Author Comment (AC1)

**Response to Reviewers:**

Thanks for the reviewer's comments on our manuscript entitled "A comprehensive study on hygroscopic behaviour and nitrate depletion of $NaNO_3$ and dicarboxylic acid mixtures: Implication for the influence factors of nitrate depletion". The reviewers' comments are helpful for improving the quality of our work. The responses to the comments and the revisions in manuscript are given point-to-point below.

**Reviewer #2:**

**Major comments**

1. My main concern is with the presentation and interpretation of several of the results with regard to the loss of nitrate from the particles/droplets via evaporation of $HNO_3$. As the title indicates, nitrate depletion is a key interest of this work, yet most of the data provided does not appear to show direct evidence for nitrate depletion via $HNO_3$ loss to the gas phase – contrary to what is stated in the abstract (line 15) and the Results and Discussion section. The measurements provided characterize the composition and spectral features of nitrate and DCA species in the condensed phase(s) of deposited particles. However, unless I missed it, $HNO_3(g)$ is not measured, nor is the mass of nitrate in the particle phase tracked. If there is substantial loss of nitrate, the particles should change in nitrate content and overall mass. Moreover, the loss of nitrate would likely be accompanied by a loss of water. As pointed out in the specific comments that follow, there are several instances where statements like "... confirms nitrate depletion and $HNO_3$ release" are made (e.g. line 133), while the information provided does not directly support this conclusion. For example, the authors seem to equate the formation of liquid or solid $NaHC_2O_4$ as evidence for nitrate depletion. However, it is not discussed why that should be a unique signature of such a process. This reviewer argues that (at least some) dissolved or crystalline $NaHC_2O_4$ could also form in the absence of any nitrate depletion, simply as a result of aqueous solution chemistry and resulting equilibria. I strongly suggest that the authors rectify this issue and provide direct evidence for $HNO_3$ loss to the gas phase or otherwise clarify that their interpretation depends on certain assumptions about the relevant processes taking place.

To be clear, I also consider nitrate depletion via $HNO_3$ to be a plausible hypothesis and a likely explanation for the findings from this work. However, what is missing is quantitative evidence and/or clarity in the discussion in support of this hypothesis.

**Author reply:** First of all, the $HCl/HNO_3$ release from mixed nitrate/chloride and organic acid particles has been widely detected in field and laboratory measurements (Laskin et al.,

2012;Wang and Laskin, 2014;Ma et al., 2019a;Ghorai et al., 2014;Shao et al., 2018). Laskin's group has employed the computer controlled scanning electron microscopy with energy dispersed analysis of X-rays (CCSEM/EDX) to measure the Cl/Na ratios in mixed organic acid/NaCl particles (Laskin et al., 2012;Ghorai et al., 2014) and N/Na ratios in mixed organic acid/$NaNO_3$ particles (Wang and Laskin, 2014). The measurement results clearly demonstrated the significant chloride/nitrate depletion by organic acids. Ma et al. (2013)

measured the water adsorption isotherms of mixed NaCl/OA particles using vapor sorption analyzer. The decreased particle mass, i.e., water content therein, indicated the formation of less hygroscopic $Na_2C_2O_4$ in internally mixed particles. The corresponding Raman spectra results also confirmed the chloride depletion. Li et al. (2017) measured the Raman features of mixed NaCl/MA particles using in situ Raman microspectrometry and found the production of monosodium malonate in mixtures during the RH cycle, which demonstrated the chloride depletion and gaseous HCl release. Furthermore, the specific reactions of strong acids displaced by weak acids in aerosols have been summarized and discussed in our preivous review (Chen et al., 2021). Second, our group has persistently utilized the FTIR spectra to characterize the chemical composition evolutions of aerosol reaction systems (Wang et al.,

2019;Shao et al., 2018;He et al., 2017;Du et al., 2020). For instance, Wang et al. (2019)

observed the ammonium depletion and gaseous $NH_3$ release from mixed dicarboxylic acid salts and $(NH_4)_2SO_4$ aerosols during the RH cycle using the ATR-FTIR technique. Similar to this work, they measured the IR feature changes of dicarboxylic acid salts and corresponding

DCAs, as well as the efflorescence transition behaviors of mixed particles. The results clearly demonstrated the compositional evolution and ammonium depletion in mixed dicarboxylic acid salts and $(NH_4)_2SO_4$ aerosols. Shao et al. (2018) determined the chemical composition evolution of MA/$NaNO_3$, MA/$Mg(NO_3)_2$ and MA/$Ca(NO_3)_2$ particles by vacuum FTIR

method. The intensity changes of feature bands of $COO^-$, $COOH$ and $NO_3^-$ groups indicated the formation of malonate salts and $HNO_3$ release. Finally, the almost unchanged water content after nitrate depletion for $DCA/NaNO_3$ mixtures in this work, e.g., $MA/NaNO_3$ mixtures in Fig. 5a in the revised manuscript, might be due to the comparable hygroscopic properties of malonate salts and MA (Jing et al., 2018;Wu et al., 2011), or limited amounts of formed dicarboxylic acid salts.

**Specific comments**

2. Title: phrasing of "Implication for the influence factors of nitrate depletion" needs language improvement and perhaps a more direct link to aerosols. Consider replacing by "implications for nitrate depletion in tropospheric aerosols".

   **Author reply:** Thanks for the reviewer's suggestion. We have adopted reviewer's advice and revised our manuscript accordingly.

3. Line 15: make statement consistent with the data provided (see my major comment).

   **Author reply:** Thanks for the reviewer's suggestion. We have illuminated the agreements on the observations of nitrate/chlorine depletion and $HNO_3/HCl$ release from mixed nitrate/chlorine and DCA particles via various measurement techniques (including FTIR spectra) from previous literatures. For clarity, we have revised "$HNO_3$ release" into "nitrate depletion" in Line 15.

4. Line 19: "... potentially indicating the transformation of amorphous solids to semisolid $NaNO_3$; ..."; consider: or the dissolution of $NaNO_3$ into a liquid DCA + Water rich phase?

   **Author reply:** As shown in Fig. S6, no obvious changes of the 1356 $cm^{-1}$ band assigned to $v_3(NO_3^-)$ of amorphous $NaNO_3$ solids were observed at constant ~ 15% RH. Meanwhile, the liquid water content remained almost unchanged as RH increased from ~ 14.8% to ~ 61.0% (seen Fig. 3b). Therefore, we can conclude the transformation of amorphous solids to semisolid $NaNO_3$, rather than the dissolution of $NaNO_3$ solids as RH increased to ~ 15%.

5. Line 39: define abbreviations like DCA, ATR-FTIR and OA at first use in the main text (aside from the abstract).

   **Author reply:** Thanks for the reviewer's suggestion. We have adopted reviewer's advice and revised our manuscript accordingly.

6.  Line 42: there are several directly relevant studies on hygroscopicity and surface tension effects that could be cited here; e.g. Peng et al. (2001, 10.1021/es0107531), Ovadnevaite et al.

(2017, doi:10.1038/nature22806), Hodas et al. (2015, doi:10.5194/acp-15-5027-2015).

**Author reply:** Thanks for the reviewer's suggestion. We have adopted the reviewer's advice and revised our manuscript accordingly.

7.  Line 43: "It is well known that the displacement of strong acids, i.e., HCl or $HNO_3$, by weak organic acids, e.g., DCAs, is not thermodynamically favoured in bulk solution". This statement requires citation of a relevant reference. Note that even if the thermodynamic equilibrium is favouring the left hand side of reaction (R1), there will be some amounts of

HCl or $HNO_3$ dissolved and if gas phase exchange is possible, there will be a continuous loss of the right hand side, consuming the reactants of R1 over possibly long time (depending on bulk vs. small aerosols). Therefore, given that in an aqueous phase, $NaNO_3$ will be present mostly in the form of $Na^+$ and $NO_3^-$, some NaA(aq) (or solid) and/or $Na^+$ and $A^-$ may be present even in the absence of any evaporation of $HNO_3$.

**Author reply:** Thanks for the reviewer's suggestion. The $HNO_3$ and HCl are the stronger acids compared to DCAs, and the displacement of weak DCAs by strong $HNO_3$ or HCl is thermodynamically favoured in bulk solutions. Contrary to this, the displacement of strong acids by weak organic acids is not thermodynamically favoured in bulk solutions. While in aerosol phases, the equilibrium of reaction R1 would shift to the right due to the efficient evaporation of HCl from the particle phase to the gas phase, in view of the much higher surface-to-volume ratios of aerosols compared to bulk solutions. Indeed, in an aqueous phase,

$NaNO_3$ would be present mostly in the form of $Na^+$ and $NO_3^-$, and HA can be dissociated into

$H^+$ and $A^-$. Whereas, not considering the displacement reaction equilibrium, the amounts of dissociated $H^+$ and $A^-$ are limited according to the acid dissociation constants of organic acids.

Based on this, the formed NaA (aqueous or solid) and dissolved HCl or $HNO_3$ are very limited and cannot be detected by our IR spectra (seen Fig. 5d and 6d). Furthermore, HCl and

$HNO_3$ have the very high volatility, i.e., low Henry's law constants (HCl: $< 2 \times 10^{-1}$; $HNO_3$: $>$

$2 \times 10^5$), and thus they tend to partition from the particle phase to the gas phase, causing the shift of the reaction R1 to the right and much more NaA formation.

8.   Line 52: State for what concentration/conditions. Are you comparing at the same RH or the same water content or the same molar concentration of dissolved solutes?

**Author reply:** Thanks for the reviewer's suggestion. We have stated that in mixed

NaCl/citric acid droplets with molar ratio of 1:1, dissociated HCl concentration is $\sim 10^{10}$

times higher than dissociated citric acid according to the different acid dissociation constants ($K_{a1}$) (HCl: $> 1 \times 10^{7}$, citric acid: $8.4 \times 10^{-4}$) (Laskin et al., 2012).

9.   Line 58: define the base of the logarithm and perhaps at first use, define the meaning of pKa1

formally using molal activities or standard chemical potentials, as appropriate given the values stated.

**Author reply:** Thanks for the reviewer's suggestion. We have adopted the reviewer's advice and revised our manuscript accordingly.

10.  Line 66: what "substrate" is here referred to? do you mean material onto which particles were deposited in laboratory experiments?

**Author reply:** The substrate materials referred to the substrates on which particles were deposited in the previous studies. For instance, in Ghorai's study, the submicrometer

NaCl/MA(GA) particles were deposited on $Si_3N_4$ windows and TEM grids (Ghorai et al.,

2014). In another study, the micron-sized $NaNO_3$/MA(GA) particles were deposited on carbon-filmed grids or $Si_3N_4$ windows (Wang and Laskin, 2014). While Ma et al. (2013)

placed mixed NaCl/MA(GA) particles in aluminum sample holder or on Ge substrate.

11.  Line 70: "... the substrate supporting OA droplets would crystallize to form"; do you mean the droplet / OA or really the substrate? Clarify phrasing.

**Author reply:** Thanks for the reviewer's suggestion. We have revised "substrate supporting

OA droplets" into "OA droplets deposited on polytetrafluoroethylene (PTFE) substrate".

12.  Line 70: "OA dihydrate at 71% RH" Is it under conditions of dehumidification? Describe what kind of laboratory experiment you refer to.

**Author reply:** Thanks for the reviewer's suggestion. We have revised "at $\sim$ 71% RH" into "at

$\sim$ 71% relative humidity (RH) during dehydration".

13.  Line 72: "Pure NaNO3 droplets might not be effloresced..." By "be effloresced" do you rather mean "crystallize" here? Note that effloresced is not necessarily the same as crystalline in the context of amorphous solids as a possibility. Also, clarify under what conditions; here presumably dehydration in an isothermal (and isobaric) experiment.

**Author reply:** Thanks for the reviewer's suggestion. "Pure $NaNO_3$ droplets might not be effloresced but convert into amorphous state at low RH" indicated that $NaNO_3$ droplets could neither be crystallized, nor be effloresced to amorphous solids, but instead formed viscous supersaturated liquids at low RH (Liu et al., 2008). For clarity, we have revised "amorphous state at low RH" into "highly concentrated droplets at low RH upon drying" in the text.

14. Line 73: "certain RH values" Which ones? Please provide values/ranges.

**Author reply:** Thanks for the reviewer's suggestion. We have adopted the reviewer's advice and revised our manuscript accordingly.

15. Line 74: Given the stated list of options, how is "highly viscous" different from glassy or semisolid? One could say "highly viscous (i.e. semisolid or glassy), liquid, or mixed-phase (e.g. solid–liquid) mixing states.

**Author reply:** Thanks for the reviewer's suggestion. We have adopted the reviewer's advice and revised our manuscript accordingly.

16. Line 75: Koop et al. (2011, doi:10.1039/C1CP22617G) would be another key reference to cite here.

**Author reply:** Thanks for the reviewer's suggestion. We have adopted the reviewer's advice and revised our manuscript accordingly.

17. Line 82: phrasing: "would" or rather "does" it?

**Author reply:** Thanks for the reviewer's suggestion. We have adopted the reviewer's advice and revised our manuscript accordingly.

18. Line 85: delete "mixture"; it is redundant with solution.

**Author reply:** Thanks for the reviewer's suggestion. We have adopted the reviewer's advice and revised our manuscript accordingly.

19. Line 92: "... a brief was introduced here"; phrasing.

**Author reply:** Thanks for the reviewer's suggestion. We have adopted the reviewer's advice and revised our manuscript accordingly.

20. Line 103: "chamber, which could be used to calibrate the ambient RH"; in the context of this description, it is mentioned that water vapor and $CO_2$ were removed by the vacuum pump. So, how was the RH established in the sample chamber? Is it due to remaining water vapor under low pressure conditions? If so, does that mean that the sample droplets are (potentially)

evaporating water alongside with $HNO_3$ and DCA? The droplets may also outgas dissolved

$CO_2$ during the experiment given the vacuum applied, which might have some effect on the acidity.

**Author reply:** Before the measurements, the water vapor and $CO_2$ were removed by the vacuum pump and the pressure in the sample chamber could arrive to $\sim 0.01$ kPa. After that, water vapor from the water reservoir was fed into the sample chamber to establish the ambient

RH in the sample chamber. When the outlet of water vapour was closed, the water vapour pressure in the sample chamber would keep equilibrium with that of water reservoir, and the maximum RH at the certain temperatures could be reached. In addition, the liquid water in the sample droplets did evaporate alongside with very small amounts of $HNO_3$ and DCAs in the vacuuming process. The $HNO_3$ release and DCA salts formation during vacuuming have been indicated in the text. Besides, the dissolved $CO_2$ in micron-sized droplets was very limited, and it was conceivable that the influence of outgassing of dissolved $CO_2$ on droplet acidity, and thus the nitrate depletion in mixed $NaNO_3$/DCA particles, was negligible.

21. Line 127: "and anhydrous OA" I suggest to mention that you refer to crystalline anhydrous

OA here.

**Author reply:** Thanks for the reviewer's suggestion. We have adopted the reviewer's advice and revised our manuscript accordingly.

22. Line 131: "... indicative of the $NaHC_2O_4$..."; Is this referring to an actual solid $NaHC_2O_4$

component forming or rather the presence of partially dissociated OA anions ($HC_2O_4^-$) in aqueous solution alongside $Na^+$? In the latter case, this should not be referred to as $NaHC_2O_4$

since it is not forming a compound of that structure nor stoichiometry. Also, in that case, this does not confirm directly the release of $HNO_3$, it only indicated the presence of partially dissociated OA. Please discuss.

**Author reply:** Thanks for the reviewer's suggestion. We have revised "$NaHC_2O_4$" into

"dissociated $HC_2O_4^-$" in the revised manuscript. As indicated above, the amounts of dissociated $HC_2O_4^-$ from OA were very limited and could not be detected by our IR spectra assuming that no displacement reactions occurred. Furthermore, the release of $HNO_3$, which has been demonstrated in the earlier studies and discussed above, would shift the reaction R1 to the right. Therefore, more OA was dissociated into $HC_2O_4^-$ in the presence of $Na^+$ along with continuous $HNO_3$ release, as indicated in reaction R2.

23. Line 132: "Likewise, Wang et al. (2017) observed the formation of $NH_4HC_2O_4$ in mixed $(NH_4)_2SO_4$/OA droplets upon drying. These scenarios confirm the nitrate depletion and $HNO_3$ release from $NaNO_3$/OA mixtures in the vacuuming process." The provided data and discussion does not uniquely support this conclusion. While $HNO_3$ evaporation is possible, where is the evidence for it? Oxalate salts could form also without any loss of $HNO_3$ to the gas phase. For example, $HNO_3$(aq) could form or, more likely, $H^+$(aq) and $NO_3^-$ (aq) alongside solid or dissolved OA salts. Please provide evidence or a more nuanced discussion.

**Author reply:** As discussed in the author reply for the major comment, the $HNO_3$ release from mixed nitrate and organic acid particles has been widely detected in field and laboratory measurements. Also, the $HNO_3$ in the aqueous phase tends to partition into the gas phase due to the high volatility, i.e., low Henry's law constants ($> 2\times10^5$), of $HNO_3$ and the high surface-to-volume ratios of aqueous droplets. Therefore, the formation of OA salts would be accompanied by $HNO_3$ release, in other words, the oxalate salts formation did confirm the nitrate depletion and $HNO_3$ release from $NaNO_3$/OA mixtures.

24. Line 145: "Likewise, Ma et al. (2013) found that the DRH of NaCl component decreased in both external and internal NaCl/MA mixtures." How could this be the case in an external mixture (assuming external means (pure) NaCl particles separate from MA particles)?

Also, worth mentioning in the context of DRH lowering: a lowering of the DRH in particles containing aqueous DCAs is expected from thermodynamic equilibrium theory and has been shown in many other measurements and by means of thermodynamic model predictions, such as those shown for MA or OA and ammonium sulfate by Bouzidi et al. (2020, doi:10.1016/j.atmosenv.2020.117481) and those shown by Hodas et al. (2015, doi:10.5194/acp-15-5027-2015) for mixtures of ammonium sulfate with DCAs and citric acid

**Author reply:** Thanks for the reviewer's suggestion. In Ma's study, they found that externally and internally mixed NaCl/MA particles exhibited DRH in the range of 65-70% RH and 60-65% RH, respectively, which were lower than the DRH of pure NaCl particles (75% RH). Indeed, it might be implausible that the dissolution of external NaCl particles were influenced by separated MA particles. One possible reason for this might be that the contact of solution films of separated NaCl and MA particles, caused by water absorption of solid particles at RH below the DRH (Bruzewicz et al., 2011;Wise et al., 2008), led to the decrease in surface tensions of solutions and enhancement of water absorption, which further reduced the DRH of pure components. Whereas for clarity, we have deleted this sentence and added the citations of relevant studies by Bouzidi et al. and Hodas et al.

25. Line 146: "Note that the OA dihydrate and crystalline $Na_2C_2O_4$ cannot be deliquesced due to their very high DRHs ...". A note: the process of deliquescence, as more clearly observed in binary (water + 1 solute) systems, should not be confused with partial (gradual) dissolution in multicomponent, multiphase systems. Therefore, please clarify the sentence. At present it is misleading. While crystalline substances like OA and $Na_2C_2O_4$ may not fully dissolve, it should be noted that in the presence of an aqueous phase a solid–liquid equilibrium will be established (given enough time) and some amount of OA and $Na_2C_2O_4$ will be in the dissolved state, forming a saturated aqueous solution with respect to the pertaining crystalline phase or phases; see, e.g. the discussion by Hodas et al. (2016, doi:10.5194/acp-16-12767-2016).

**Author reply:** Thanks for the reviewer's suggestion. We have adopted the reviewer's advice and revised our manuscript accordingly.

26. Line 155: "Upon hydration, the water content at high RH is far below that upon dehydration, ..." This statement needs further clarification. It seems only to make sense when the dehydration refers to dehydration from an initial state of completely dissolved particles at very high RH. Of course, one could start dehydration at any RH and then the water content could be higher or lower than when arrived at that state from an initial state of low RH.

**Author reply:** Thanks for the reviewer's suggestion. We have adopted the reviewer's advice and revised our manuscript accordingly.

27. Line 159: I find Fig. S6 insightful and suggest that this figure and the related discussion be moved to the main manuscript.

**Author reply:** Thanks for the reviewer's suggestion. We have adopted the reviewer's advice and revised our manuscript accordingly.

28. Line 160: "Besides, as shown in Fig. 1c, the absorbance of 1620 cm$^{-1}$ band assigned to oxalate shows a slight increase at RH as low as 21.2%, implying the nitrate depletion proceeds at relatively low RH."

In my opinion, what is missing are observations of the loss of nitrate/HNO$_3$ from the particles.

Do the authors have comparison examples for cases where the gas phase exchange is limited (small gas volume) such that only a small loss could occur and, as a result, the signature in oxalate formation substantially different? Without such observations, or other quantitative measures of loss to the gas phase, the interpretation of the experiments hinges on assuming a nitrate depletion takes place without having the data to confirm it. Please discuss.

**Author reply:** As already indicated, Li et al. (2017) observed the Raman features of monosodium malonate in NaCl/MA mixed aerosols during the RH cycle, indicating the chloride depletion and gaseous HCl release. Wang et al. (2019) confirmed the ammonium depletion and gaseous NH$_3$ release from mixed dicarboxylic acid salts and $(NH_4)_2SO_4$

aerosols during the RH cycle according to the IR feature changes of dicarboxylic acid salts and corresponding DCAs via the ATR-FTIR technique. Ma et al. (2013) also determined the oxalate formation and gaseous HCl release from NaCl/OA mixtures based on the Raman and

ATR-FTIR characterization results. Therefore, the increase in absorbance of IR feature bands of oxalate is proved to be powerful evidence for nitrate depletion and gaseous HNO$_3$ release.

29. Line 173: "As RH increases to ~ 15%, the $\Delta A$ of $\nu_{as}$(COO-) band exhibits a considerable increase, indicative of the occurrence of nitrate depletion."

While indicative as an option, it remains speculative. It is indicative of the partial dissociation of OA in solution. Which is expected to increase with increasing particle water content and, hence, increasing RH.

**Author reply:** As already indicated, there were no IR feature bands of COO$^-$, e.g., 1465 cm$^{-1}$

band, in IR spectra of pure OA droplets, as shown in Fig. S3. Thus, the increase in $\Delta A$ of

$\nu_{as}$(COO$^-$) band did indicate the occurrence of nitrate depletion.

30. Line 177: "2013). Then, the 1620 cm$^{-1}$ band appears and becomes stronger with time, suggesting $Na_2C_2O_4$ can be continuously produced at constant $\sim 15\%$ RH."

I think this is better evidence for $Na_2C_2O_4$ formation and associated nitrate depletion, because the system is observed at constant RH, which means it should maintain constant water activity (in sufficiently large particles where the Kelvin effect would not change with evaporation). It may be good to expand the discussion of this piece of evidence.

**Author reply:** Thanks for the reviewer's suggestion. We have adopted the reviewer's advice and revised our manuscript accordingly.

31. Line 179: "... we can infer the conversion of amorphous $NaNO_3$ solids to highly viscous semisolids due to the uptake of trace amounts of moisture."

Please clarify: semisolids of what? if $NaNO_3$ dissolves partially and forms a viscous aqueous solution, this should not be called a $NaNO_3$ semisolid because the formed phase contains other species too.

**Author reply:** Thanks for the reviewer's suggestion. We inferred that amorphous $NaNO_3$

solids did not dissolve partially, based on the IR feature changes of $NO_3^-$ as discussed above, but transformed into highly viscous $NaNO_3$ semisolids. We have adopted the reviewer's advice and revised our manuscript accordingly.

32. Line 216: Define the $R^2$ metric.

**Author reply:** Thanks for the reviewer's suggestion. We have adopted the reviewer's advice and revised our manuscript accordingly.

33. Line 231: "As RH increases to around 15%, amorphous solids are converted into viscous semisolids, ...". Is there any evidence from the spectra for such a conversion and the approximate viscosity of the phase to classify it as semisolid?

**Author reply:** Thanks for the reviewer's suggestion. There were no spectra and viscosity data for the conversion of amorphous solids to semisolids. First of all, $NaNO_3$ has been proved to exist in unusual metastable states, e.g., amorphous solids or highly concentrated droplets, after drying (Ma et al., 2021;Liu et al., 2008;Hoffman et al., 2004). Then, the chemical reactivity to liquid OA did alter as RH increased to $\sim 15\%$ herein. Coupled with the particle morphology changes shown in Fig. 2, we can speculate that amorphous $NaNO_3$ solids were converted into viscous NaNO$_3$ semisolids as RH increased to ~ 15%.

34. Line 261: "... formation of monosodium malonate ...". Is this for a solid malonate phase or dissolved?

**Author reply:** Li et al. (2017) measured the Raman spectra of pure monosodium malonate (MSM) aerosols at RH = 10% and 2:1 NaCl/MA mixed aerosols at very low RH of 9.1% upon dehydration, which did not resemble that of MSM powders, suggesting that the pure MSM

aerosols and mixed NaCl/MA aerosols tended to form amorphous states after drying.

Furthermore, in this work, the IR feature band of MSM, i.e., 1595 cm$^{-1}$ band, showed no obvious changes during humidification, as shown in Fig. 5c, indicating no deliquescence transition of MSM occurrence. Based on these, we can infer that the MSM component in mixed NaNO$_3$/MA aerosols was in amorphous liquids, rather than solid phases.

35. Line 266: Clarify: "that the first acid dissociation constant (pKa1) of MA was about 2.83, which was ~ 3 orders of magnitude larger than the second one ...". 3 orders of magnitude in what? clearly not in pKa. Also, 2.83 is the pKa1 not Ka1 (which the current phrasing implies).

**Author reply:** Thanks for the reviewer's suggestion. We have adopted the reviewer's advice and revised our manuscript accordingly.

36. Line 270: band assigned to $\nu_{as}$(COO$^-$) of monosodium malonate. Please clarify: why of monosodium malonate and not of the malonate anion? The COO$^-$ functionality does not contain any sodium and in aqueous solution the malonate ion is likely dissociated from Na$^+$. Is the assigned band specific to the COO$^-$ group in crystalline monosodium malonate only?

**Author reply:** Thanks for the reviewer's suggestion. We have adopted the reviewer's advice and revised our manuscript accordingly.

37. Line 277: "aqueous NaNO$_3$" should be "aqueous NO$_3$.

**Author reply:** Thanks for the reviewer's suggestion. We have adopted the reviewer's advice and revised our manuscript accordingly.

38. Line 279: "As compared to vacuum FTIR results, we can infer that the heterogeneous efficacy of Ge substrate is much higher than CaF$_2$ windows,...". It seems unclear whether this is really a substrate effect or rather an effect of the vacuum cell procedures applied

**Author reply:** In our previous review, the efflorescence kinetics and nucleation mechanisms were discussed detailed. The substrate effects have been proved to be a key factor to induce heterogeneous nucleation of aerosol particles. Furthermore, the heterogeneous nucleation efficiency of $CaF_2$, Ge and ZnSe substrates for deposited particles have been demonstrated in our earlier studies (Ma et al., 2019b;Ji et al., 2017;Zhang et al., 2014;Ren et al., 2016).

39. Line 290, Fig. 5: What explains the difference in the normalized water contents comparing panels (a) and (b)? Did you compare the obtained water contents to model predictions of the water content or independent mass growth factor measurements to assess which technique provides more accurate hygroscopic cycles?

**Author reply:** The difference in the normalized water content in panels (a) and (b) was attributed to the different sequences of water cycles. In other words, the deposited droplets in vacuum FTIR were first dried in vacuum, and then underwent a humidification-dehumidification cycle. According to the weak absorbance intensity of 1548

$cm^{-1}$ band shown in Fig. 6c in the revised manuscript, we can infer that the nitrate depletion was limited and might not alter the hygroscopic ability of mixed aerosols. While for

ATR-FTIR measurement, the mixed droplets first underwent the dehumidification process and then a humidification process. Upon hydration, the GA component mainly existed in solid state and could not be dissolved completely even until 84% RH, and thus the water content could not match with that in the dehumidification process. Note that the complete deliquescence of mixed $NaNO_3$/GA particles in vacuum FTIR measurement might be due to the partial crystallization of GA in the vacuuming process.

Firstly, the formation and precipitation of organic acid salts have not been considered by the current version of E-AIM, thus, the model cannot provide accurate hygroscopic growth factors for $NaNO_3$/DCA mixtures. Second, the mixed aerosols (e.g., $NaNO_3$/OA mixtures)

might exist in solid-liquid mixing state, not the aqueous solution state at high RH in the vacuum FTIR measurement. Therefore, we could not determine the mass growth factors of mixed particles through the absorbance of liquid water band and E-AIM predictions (even though the data was not accurate), according to the calculation method proposed by our earlier studies (Ma et al., 2019b;Ji et al., 2017). Note that the mass growth factors and E-AIM

predictions of pure $NaNO_3$ and DCA aerosols have been determined and shown in the

Supplement.

40. Line 301: "appears upon hydration, indicating the formation of glutarate sodium salts." Could you clarify whether this means that a solid crystalline salt phase is forming upon hydration in equilibrium with an aqueous solution?

**Author reply:** Thanks for the reviewer's suggestion. According to the hygroscopic behaviours of mixed $NaNO_3$/GA particles and IR feature changes of $\nu_{as}(COO^-)$, which was similar to that of mixed $NaNO_3$/MA particles, we can infer that the formed glutarate sodium salts were present in aqueous solution state. Furthermore, we have revised our manuscript accordingly.

41. Line 314: "lower temperature of droplets" Is it just the temperature effect or also the drying to much lower effective RH than during the ATR-FTIR measurement cycles?

**Author reply:** The lowest RH in the two measurements was comparable, i.e., 3.4% RH in vacuum FTIR vs. 5.6% RH in ATR-FTIR. The temperature effect caused by rapid water evaporation has been introduced in our previous study, which can facilitate the nucleation of aqueous droplets (Ma et al., 2019b).

42. Line 335: Could you discuss in this context whether the size dependence effect is due to the

Kelvin effect, leading to a lower water activity and higher pH in smaller particles compared to larger ones exposed to the same RH in the gas phase, or some other effect?

**Author reply:** Thanks for the reviewer's suggestion. As indicated in our previous review, the depletion extent, $\xi$, was related to depleted mass ($\Delta m$) and initial mass ($m_0$) of a volatile species within a droplet with the radius $r$, which could be expressed as $\xi = \Delta m/m_0$ (Chen et al.,

2021). The $\Delta m$ could be calculated by $\Delta m = 4\pi r MDp_r/(RT\Delta t)$ via the Maxwell steady-state diffusive mass transfer equation, where $M$, $D$, and $p_r$ denoted the volatile species molecular weight, diffusion coefficient in the air, and equilibrium partial pressure at the droplet surface, respectively; $R$, $T$, and $\Delta t$ were the ideal gas constant, temperature, and reaction time, respectively. The $m_0$ of the volatile species could be expressed as $m_0 = 4\pi r^3 c/3$, since the concentration of the volatile species ($c$) within the droplet kept constant at a fixed RH. Based on these, the $\xi$ could be expressed as

$$\xi = \frac{k}{r^2}\Delta t, \quad \text{with } k = \frac{3MDp_r}{4\pi cRT} \tag{1}$$

As seen, the depletion extent, $\xi$, was inversely proportional to $r^2$. Therefore, the size dependence effect of depletion extent was mainly due to the volatility difference, rather than the Kelvin effect.

43. Line 346, Eq. (5): check equation; it looks like the $p$ of $p_r$ is missing.

**Author reply:** Thanks for the reviewer's suggestion. We have adopted the reviewer's advice and revised our manuscript accordingly.

44. Line 358: Eq. (6), for consistency with Eq. (7), consider replacing $P$ by $p*$.

**Author reply:** Thanks for the reviewer's suggestion. Firstly, we indicated the expression of the diffusion coefficient of $HNO_3$ in the gas phase according to Chapman-Enskog method. In other words, the $D$ denoted to the diffusion coefficient of $HNO_3$ in the ATR-FTIR

measurement. Then, the $HNO_3$ diffusion coefficient in vacuum FTIR measurement, which was defined as $D*$, was related to the ambient pressure in vacuum FTIR, $P*$. Thus, the $P$ in the eq. (6) did not only denote to $P*$ in the eq. (7).

45. Line 373: "Therefore, the ambient pressure, dominating the diffusion coefficient of $HNO_3$ in the gas phase, tends to play an important role in nitrate depletion during the transport and aging of atmospheric aerosols."

This statement is not supported by any quantitative data. How big of a difference is expected in the troposphere? Obviously, one should not directly compare the vaccum FTIR pressure to realistic tropospheric conditions without accounting for the differences in the ranges of total pressures. If nitrate-containing particles are given hours to days to undergo displacement reactions, the impact of pressure on $HNO_3$ diffusion may be unimportant. Do you have predictions to the contrary?

**Author reply:** Thanks for the reviewer's constructive suggestion. We have deleted the improper statement, i.e., "Therefore, the ambient pressure, dominating the diffusion coefficient of $HNO_3$ in the gas phase, tends to play an important role in nitrate depletion during the transport and aging of atmospheric aerosols." in the revised manuscript.

46. Line 409: "NaNO3 can be treated as a surrogate for a broad class of amorphous or semisolid species existing in atmospheric aerosols,..."

Please state in what RH range this semisolid state applies. NaNO3 is certainly dissolved and liquid-like in dilute aqueous aerosols at elevated RH (for aqueous NaNO3 viscosity data see e.g. Baldelli et al., 2016, doi:10.1080/02786826.2016.1177163; Lilek and Zuend, 2022, doi:10.5194/acp-22-3203-2022).

**Author reply:** Thanks for the reviewer's suggestion. We have adopted the reviewer's advice and revised our manuscript accordingly.

47. Line 427: "metastable or liquid-like state..." Please rephrase; metastable and liquid-like characterize two different properties. One related to the supersaturation of aqueous solutions with respect to a certain crystalline phase, while the other refers to a phase state (viscosity related).

**Author reply:** Thanks for the reviewer's suggestion. We have adopted the reviewer's advice and revised our manuscript accordingly.

**Supplement: minor comments**

48. SI, lines 20–36: How was that ERH determined? Was it based on just some features of the spectra? Fig. S1a does not seem to indicate any clear signature of efflorescence. Specifically, note that in the absence of severe kinetic limitations, one would expect a sharp change in mass growth factor at the ERH (point of crystallization) of a single-solute particle. Certainly so for a system at equilibrium with the gas phase. Why is that not the case for the systems shown in

Fig. S1? Figure S2 shows also no indication of NaNO3 crystallization. Do the authors use the term "efflorescence" for a phase transition other than crystallization? If so, that would be untypical given the common use and meaning of this term and might need clarification in the text.

**Author reply:** The ERH of $NaNO_3$ was determined via the IR feature changes of $NO_3^-$, i.e., the broad ~ 1350 cm$^{-1}$ band was transformed into a sharper peak, as shown in Fig. S1b and

S2b. The absence of sharp decrease in mass growth factor at the ERH was attributed to the specific structure of nitrates after drying, i.e., amorphous solids. Likewise, Li et al. (2021)

observed the hygroscopic behaviours of pure $(NH_4)_2SO_4$ and $NH_4NO_3$ particles during the RH

cycles. They found that mass growth factor of $(NH_4)_2SO_4$ particles would decrease sharply at the ERH, while mass growth factor of $NH_4NO_3$ particles decreased routinely without the turning point at the ERH. Meanwhile, the 1435 cm$^{-1}$ band assigned to $NH_4^+$ in the solution phase red-shifted to 1415 cm$^{-1}$, suggesting the occurrence of efflorescence of $NH_4NO_3$

particles at 25% RH. Furthermore, Tang and Fung (1997) found that Raman spectra characteristic of dried $Ca(NO_3)_2$ particles was obviously different from that of anhydrous crystals, indicating the amorphous solids formation. Meanwhile, no sharp decrease in the water content of $Ca(NO_3)_2$ particles was observed during the dehumidification, and the amorphous solids would deliquesce at RH well below the DRH of anhydrous crystals, which showed good agreement with the observations for $NaNO_3$ particles in this work. In addition, the term "efflorescence" did indicate the formation of amorphous solids rather than the crystallization here. For clarity, we have revised our manuscript accordingly.

49. SI, line 40: what about that of a $NaNO_3$ hydrate crystal? Would equilibrium thermodynamics not require that amorphous solids take up some water upon hydration (while single-solute crystalline particles may not)?

**Author reply:** As indicated in the Supplement, the $v_3(NO_3^-)$ feature bands after drying in the two measurements are inconsistent with the IR feature of $NaNO_3$ crystals, i.e., two shoulder peaks centred at 1383 and 1485 cm$^{-1}$ arising from splitting of the degenerate $v_3$ mode (Liu et al., 2008). Liu et al. (2008) also found small amounts of water present within particles after dehydration and the particles absorb water continuously upon hydration, indicating the formation of highly concentrated droplets after drying, rather than amorphous $NaNO_3$ solids.

The amorphous nitrate solids (e.g., $Sr(NO_3)_2$ and $Ca(NO_3)_2$) would not obviously absorb water until their DRHs (Tang and Fung, 1997).

50. SI, lines 155–158, 166: these sentences are inconsistent. One is about HCl release, the other about $HNO_3$ but referring to the same study. Please clarify.

**Author reply:** Thanks for the reviewer's suggestion. We have revised the error in the revised manuscript.

51. SI, line 178: molecular weight (or rather molar mass) of water should be stated in kg mol$^{-1}$ to stay consistent with the other units given. Other there will be an incorrect Kelvin effect value.

**Author reply:** Thanks for the reviewer's suggestion. We have revised the error in the revised manuscript.

52. SI, line 188: It should be stated that the true equilibria in aqueous solution involve use of the activities of the species, not the molar concentrations. Also, on line 210, state that self-dissociation of $H_2O$ is not considered.

**Author reply:** Thanks for the reviewer's suggestion. The calculated pH here can be defined as free-$H^+$ approximation of pH on a molality basis, as indicated by Pye et al. (2020).

Furthermore, we have considered the self-dissociation of $H_2O$ in eq. (11) in the revised

Supplement.

53. SI, line 216: given definition of pH; note that this refers only to an approximate, molarity-based pH, as discussed by Pye et al. (2020, doi:10.5194/acp-20-4809-2020). The proper definition of pH involves the molality-based activity of $H^+$.

**Author reply:** Thanks for the reviewer's suggestion. We have adopted the reviewer's advice and revised our manuscript accordingly.

Reference:

Bruzewicz, D. A., Checco, A., Ocko, B. M., Lewis, E. R., McGraw, R. L., and Schwartz, S. E.:
Reversible uptake of water on NaCl nanoparticles at relative humidity below deliquescence point
observed by noncontact environmental atomic force microscopy, J. Chem. Phys., 134, 044702,
https://doi.org/10.1063/1.3524195, 2011.
Chen, Z., Liu, P., Liu, Y., and Zhang, Y. H.: Strong acids or bases displaced by weak acids or bases in
aerosols: Reactions driven by the continuous partitioning of volatile products into the gas phase, Acc.
Chem. Res., 54, 3667-3678, https://doi.org/10.1021/acs.accounts.1c00318, 2021.
Du, C. Y., Yang, H., Wang, N., Pang, S. F., and Zhang, Y. H.: pH effect on the release of $NH_3$ from the
internally mixed sodium succinate and ammonium sulfate aerosols, Atmos. Environ., 220, 117101,
https://doi.org/10.1016/j.atmosenv.2019.117101, 2020.
Ghorai, S., Wang, B. B., Tivanski, A., and Laskin, A.: Hygroscopic properties of internally mixed
particles composed of NaCl and water-soluble organic acids, Environ. Sci. Technol., 48, 2234-2241,
https://doi.org/10.1021/es404727u, 2014.
He, X., Leng, C., Pang, S., and Zhang, Y.: Kinetics study of heterogeneous reactions of ozone with
unsaturated fatty acid single droplets using micro-FTIR spectroscopy, RSC Adv., 7, 3204-3213,
https://doi.org/10.1039/c6ra25255a, 2017.
Hoffman, R. C., Laskin, A., and Finlayson-Pitts, B. J.: Sodium nitrate particles: physical and chemical
properties during hydration and dehydration, and implications for aged sea salt aerosols, J. Aerosol Sci.,
35, 869-887, https://doi.org/10.1016/j.jaerosci.2004.02.003, 2004.
Ji, Z. R., Zhang, Y., Pang, S. F., and Zhang, Y. H.: Crystal nucleation and crystal growth and mass
transfer in internally mixed sucrose/$NaNO_3$ particles, J. Phys. Chem. A, 121, 7968-7975,
https://doi.org/10.1021/acs.jpca.7b08004, 2017.
Jing, B., Wang, Z., Tan, F., Guo, Y. C., Tong, S. R., Wang, W. G., Zhang, Y. H., and Ge, M. F.:
Hygroscopic behavior of atmospheric aerosols containing nitrate salts and water-soluble organic acids,

Atmos. Chem. Phys., 18, 5115-5127, https://doi.org/10.5194/acp-18-5115-2018, 2018.

Laskin, A., Moffet, R. C., Gilles, M. K., Fast, J. D., Zaveri, R. A., Wang, B. B., Nigge, P., and Shutthanandan, J.: Tropospheric chemistry of internally mixed sea salt and organic particles: Surprising reactivity of NaCl with weak organic acids, J. Geophys. Res.-Atmos., 117, D15302, https://doi.org/10.1029/2012jd017743, 2012.

Li, Q., Ma, S. S., Pang, S. F., and Zhang, Y. H.: Measurement on mass growth factor of $(NH_4)_2SO_4$, $NH_4NO_3$ and mixed $(NH_4)_2SO_4/NH_4NO_3$ aerosols under linear RH changing mode, Spectrosc. Spectral Anal., 41, 3444-3450, https://doi.org/10.3964/j.issn.1000-0593(2021)11-3444-07, 2021.

Li, X., Gupta, D., Lee, J., Park, G., and Ro, C.-U.: Real-time investigation of chemical compositions and hygroscopic properties of aerosols generated from NaCl and malonic acid mixture solutions using in situ Raman microspectrometry, Environ. Sci. Technol., 51, 263-270, https://doi.org/10.1021/acs.est.6b04356, 2017.

Liu, Y., Yang, Z. W., Desyaterik, Y., Gassman, P. L., Wang, H., and Laskin, A.: Hygroscopic behavior of substrate-deposited particles studied by micro-FT-IR spectroscopy and complementary methods of particle analysis, Anal. Chem., 80, 633-642, https://doi.org/10.1021/ac701638r, 2008.

Ma, Q. X., Ma, J. Z., Liu, C., Lai, C. Y., and He, H.: Laboratory study on the hygroscopic behavior of external and internal $C_2$–$C_4$ dicarboxylic acid–NaCl mixtures, Environ. Sci. Technol., 47, 10381-10388, https://doi.org/10.1021/es4023267, 2013.

Ma, Q. X., Zhong, C., Liu, C., Liu, J., Ma, J. Z., Wu, L. Y., and He, H.: A comprehensive study about the hygroscopic behavior of mixtures of oxalic acid and nitrate salts: Implication for the occurrence of atmospheric metal oxalate complex, ACS Earth Space Chem., 3, 1216-1225, https://doi.org/10.1021/acsearthspacechem.9b00077, 2019a.

Ma, S. S., Yang, W., Zheng, C. M., Pang, S. F., and Zhang, Y. H.: Subsecond measurements on aerosols: From hygroscopic growth factors to efflorescence kinetics, Atmos. Environ., 210, 177-185, https://doi.org/10.1016/j.atmosenv.2019.04.049, 2019b.

Ma, S. S., Yang, M., Pang, S. F., and Zhang, Y. H.: Subsecond measurement on deliquescence kinetics of aerosol particles: Observation of partial dissolution and calculation of dissolution rates, Chemosphere, 264, 128507, https://doi.org/10.1016/j.chemosphere.2020.128507, 2021.

Pye, H. O. T., Nenes, A., Alexander, B., Ault, A. P., Barth, M. C., Clegg, S. L., Collett Jr, J. L., Fahey, K. M., Hennigan, C. J., Herrmann, H., Kanakidou, M., Kelly, J. T., Ku, I. T., McNeill, V. F., Riemer, N., Schaefer, T., Shi, G., Tilgner, A., Walker, J. T., Wang, T., Weber, R., Xing, J., Zaveri, R. A., and Zuend, A.: The acidity of atmospheric particles and clouds, Atmos. Chem. Phys., 20, 4809-4888, https://doi.org/10.5194/acp-20-4809-2020, 2020.

Ren, H. M., Cai, C., Leng, C. B., Pang, S. F., and Zhang, Y. H.: Nucleation kinetics in mixed $NaNO_3$/glycerol droplets investigated with the FTIR-ATR technique, J. Phys. Chem. B, 120, 2913-2920, https://doi.org/10.1021/acs.jpcb.5b12442, 2016.

Shao, X., Wu, F. M., Yang, H., Pang, S. F., and Zhang, Y. H.: Observing $HNO_3$ release dependent upon metal complexes in malonic acid/nitrate droplets, Spectrochim. Acta A, 201, 399-404, https://doi.org/10.1016/j.saa.2018.05.026, 2018.

Tang, I. N., and Fung, K. H.: Hydration and Raman scattering studies of levitated microparticles: $Ba(NO_3)_2$, $Sr(NO_3)_2$, and $Ca(NO_3)_2$, J. Chem. Phys., 106, 1653-1660, https://doi.org/10.1063/1.473318, 1997.

Wang, B. B., and Laskin, A.: Reactions between water-soluble organic acids and nitrates in atmospheric aerosols: Recycling of nitric acid and formation of organic salts, J. Geophys. Res.-Atmos.,

119, 3335-3351, https://doi.org/10.1002/2013jd021169, 2014.

Wang, N., Jing, B., Wang, P., Wang, Z., Li, J. R., Pang, S. F., Zhang, Y. H., and Ge, M. F.:

Hygroscopicity and compositional evolution of atmospheric aerosols containing water-soluble carboxylic acid salts and ammonium sulfate: Influence of ammonium depletion, Environ. Sci. Technol.,

53, 6225-6234, https://doi.org/10.1021/acs.est.8b07052, 2019.

Wise, M. E., Martin, S. T., Russell, L. M., and Buseck, P. R.: Water uptake by NaCl particles prior to deliquescence and the phase rule, Aerosol Sci. Technol., 42, 281-294, https://doi.org/10.1080/02786820802047115, 2008.

Wu, Z. J., Nowak, A., Poulain, L., Herrmann, H., and Wiedensohler, A.: Hygroscopic behavior of atmospherically relevant water-soluble carboxylic salts and their influence on the water uptake of ammonium sulfate, Atmos. Chem. Phys., 11, 12617-12626, https://doi.org/10.5194/acp-11-12617-2011,

2011.

Zhang, Q. N., Zhang, Y., Cai, C., Guo, Y. C., Reid, J. P., and Zhang, Y. H.: In situ observation on the dynamic process of evaporation and crystallization of sodium nitrate droplets on a ZnSe substrate by

FTIR-ATR, J. Phys. Chem. A, 118, 2728-2737, https://doi.org/10.1021/jp412073c, 2014.

---

## Author Comment (AC2)

**Response to Reviewers:**

Thanks for the reviewer's comments on our manuscript entitled "A comprehensive study on hygroscopic behaviour and nitrate depletion of $NaNO_3$ and dicarboxylic acid mixtures:

Implication for the influence factors of nitrate depletion". The reviewers' comments are helpful for improving the quality of our work. The responses to the comments and the revisions in manuscript are given point-to-point below.

**Reviewer #1:**

1.  In the abstract, the authors suggested that "The $HNO_3$ release from $NaNO_3$/OA mixtures was observed in both the measurements, owing to the relatively high acidity of OA". What does

"the relatively high acidity of OA" mean? Compared with MA and GA, or $HNO_3$? This should be revised to avoid misunderstanding.

**Author reply:** Thanks for the reviewer's suggestion. The relatively high acidity of OA

indicated that the acidity of OA was higher than MA and GA, but lower than $HNO_3$. Indeed, this expression is a bit misleading. Thus, the sentence "The $HNO_3$ release from $NaNO_3$/OA

mixtures was observed in both the measurements, owing to the relatively high acidity of OA"

has been revised to "The $HNO_3$ release from $NaNO_3$/OA mixtures was observed in both the measurements, owing to the relatively higher acidity of OA compared to MA and GA" in the revised manuscript.

2.  Line 37: Is there considerable amounts of nitrates present in sea salt aerosols? Or did the nitrate depletion frequently occur in sea salt aerosols?

**Author reply:** As known, nitrogen oxides in the atmosphere can undergo heterogeneous reactions with NaCl on the highly reactive surfaces of sea salt particles. The proposed reaction pathways are as follows (Gibson et al., 2006).

$$NaCl(s, aq) + HNO_3(g) \rightarrow NaNO_3 \ (s, aq) + HCl(g) \qquad \text{(R1)}$$

$$NaCl(s, aq) + N_2O_5(g) \rightarrow NaNO_3 \ (s, aq) + ClNO_2(g) \qquad \text{(R2)}$$

$$NaCl(s, aq) + 2NO_2(g) \rightarrow NaNO_3 \ (s, aq) + ClNO(g) \qquad \text{(R3)}$$

Furthermore, the heterogeneous uptake coefficient ($\gamma$) of $HNO_3$ and $N_2O_5$ on NaCl particles has been measured to be about $(1.0 \pm 0.8) \times 10^{-3}$ and $(2.9 \pm 1.7) \times 10^{-3}$, respectively (Hoffman et al., 2003a;Hoffman et al., 2003b). Thus, the chloride in sea salt aerosols can be substituted significantly by nitrate, as observed in field measurements by Kerminen et al.

(1998). Coupled with the gas-particle partitioning of dicarboxylic acids, we can infer that nitrate depletion by DCAs would occur frequently in sea salt aerosols.

3.  The authors observed the $NaHC_2O_4$ formation in $NaNO_3$/OA mixed system, and then

$NaHC_2O_4$ was transformed into $Na_2C_2O_4$ with further nitrate depletion. Whereas, MA was found to produce monosodium malonate as it reacted with nitrate. Why? Please clarify it in the text.

**Author reply:** Thanks for the reviewer's suggestion. The OA has a $pK_{a2}$ of 4.19, which is lower than that of MA ($pK_{a2}$ = 5.70). This implied that more $C_2O_4^{2-}$ were dissociated from OA

than $CH_2(COO)_2^{2-}$ from MA. Thus, the disodium oxalate was mainly formed in $NaNO_3$/OA

system, while monosodium malonate was the main product as MA reacted with $NaNO_3$. We have added the sentence "Besides, no disodium salts are observed in the $NaNO_3$/MA system, differing from the $NaNO_3$/OA system, which may be due to the higher acidity of OA ($pK_{a2}$ =

4.19) than MA ($pK_{a2}$ = 5.70)." in Line 270.

4.  Line 132: "Likewise, Wang et al. (2017) observed the formation of $NH_4HC_2O_4$ in mixed

$(NH_4)_2SO_4$/OA droplets upon drying." So what's the similarity of Wang's study and this work?

Did they propose the similar driving force for $HC_2O_4^-$ ions formation?

**Author reply:** Wang et al. (2017) has indicated that OA could react with $(NH_4)_2SO_4$ to form

$NH_4HSO_4$ and $NH_4HC_2O_4$ via the following pathway:

$$H_2C_2O_4(aq) \rightarrow H^+(aq) + HC_2O_4^-(aq) \qquad \text{(R4)}$$

$$H^+(aq) + SO_4^{2-}(aq) \rightarrow HSO_4^-(aq) \qquad \text{(R5)}$$

$$2NH_4^+(aq) + HSO_4^-(aq) + HC_2O_4^-(aq) \rightarrow NH_4HSO_4(aq) + NH_4HC_2O_4(aq) \qquad \text{(R6)}$$

In the first step, the OA ($pK_{a1}$ = 1.23) was dissociated into $HC_2O_4^-$ and $H^+$ ions. Then, the $H^+$

reacted with $SO_4^{2-}$ to form $HSO_4^-$ ions. While in this work, the dissociated $H^+$ ions would interact with $NO_3^-$ to produce gaseous $HNO_3$, causing nitrate depletion. After that, the $HC_2O_4^-$

ions further combined with $NH_4^+$ to produce $NH_4HC_2O_4$ in Wang's work, and in this study, the $HC_2O_4^-$ ions tended to combine with $Na^+$ to generate $NaHC_2O_4$. Besides, the formation of both $NH_4HC_2O_4$ and $NaHC_2O_4$ was observed in the drying process in the two studies.

Therefore, We proposed that "Likewise, Wang et al. (2017) observed the formation of

$NH_4HC_2O_4$ in mixed $(NH_4)_2SO_4$/OA droplets upon drying.".

5. Line 143: As indicated in this work, $NaNO_3$ deliquescence proceeds at 46.9%-61.9% RH for

$NaNO_3$/OA mixtures and ~65%-77% RH for pure $NaNO_3$. This implied that $NaNO_3$ solids began to deliquesce at RH significantly lower than its DRH. So is the deliquescence process a thermodynamic process or a kinetic process?

**Author reply:** In our previous work, the $NaNO_3$ solids were found to begin to dissolve at

~63.70% RH during the humidification, which was well below the predicted DRH by EAIM

model (Ma et al., 2021). Meanwhile, the deliquescence behavior of inorganic salt particles (i.e., NaCl, $NaNO_3$ and $K_2CO_3$) has been proved to be a dynamic process, i.e., the particles would absorb water to form the partially dissolved phase state at RH lower than their DRH, and then they were deliquesced completely once the DRH was reached (Bruzewicz et al.,

2011;Esat et al., 2018;Ma et al., 2021).

6. Line 168: The authors observed phase transition and nitrate depletion of 1:1 $NaNO_3$/OA

mixtures in Sec. 3.1, but why did they choose 3:1 mixtures to further investigate the phase state effect?

**Author reply:** As indicated in this work, the final reaction product of $NaNO_3$/OA system was

$Na_2C_2O_4$. Thus, we chose 3:1 $NaNO_3$/OA mixtures, in which the amount of reactant $NaNO_3$

was in excess according to stoichiometry, to examine whether the liquid OA could be consumed completely by aqueous $NaNO_3$. Based on this, we indicated that "At ~ 69.2% RH, the presence of 1741 $cm^{-1}$ band indicates the excess of liquid OA, suggesting this displacement reaction tends to reach equilibrium with comparable final concentrations of

"reactants" and "products".".

7. Line 231: The 15% RH was only a preset RH value with stepwise increasing RH, so it should be revised to "As RH increases to around 15% (or even lower)".

**Author reply:** Thanks for the reviewer's suggestion. We have adopted reviewer's advice and revised our manuscript accordingly.

8. Line 287: The authors also observed the chlorine depletion in 1:1 mixed NaCl/MA particles with two different RH changing rates, but a brief discussion about the experimental results should be presented in the text.

**Author reply:** Thanks for the reviewer's suggestion. We have adopted reviewer's advice and revised our manuscript accordingly.

9.   Line 413-424: I suggest that this paragraph should be rewrote to better illustrate the atmospheric implications of the experimental observations in view of the presence of mineral dust inclusions and so on in atmospheric aerosols, which constantly induce the heterogeneous nucleation of aerosols at relatively high RH.

**Author reply:** Thanks for the reviewer's suggestion. We have adopted reviewer's advice and rewrote this paragraph to illustrate the atmospheric implications of the influence of efflorescence behaviors of aerosols on nitrate depletion.

Reference:

Bruzewicz, D. A., Checco, A., Ocko, B. M., Lewis, E. R., McGraw, R. L., and Schwartz, S. E.:
Reversible uptake of water on NaCl nanoparticles at relative humidity below deliquescence point
observed by noncontact environmental atomic force microscopy, J. Chem. Phys., 134, 044702,
https://doi.org/10.1063/1.3524195, 2011.
Esat, K., David, G., Poulkas, T., Shein, M., and Signorell, R.: Phase transition dynamics of single
optically trapped aqueous potassium carbonate particles, Phys. Chem. Chem. Phys., 20, 11598-11607,
https://doi.org/10.1039/c8cp00599k, 2018.
Gibson, E. R., Hudson, P. K., and Grassian, V. H.: Physicochemical properties of nitrate aerosols:
Implications for the atmosphere, J. Phys. Chem. A, 110, 11785-11799,
https://doi.org/10.1021/jp063821k, 2006.
Hoffman, R. C., Gebel, M. E., Fox, B. S., and Finlayson-Pitts, B. J.: Knudsen cell studies of the
reactions of $N_2O_5$ and $ClONO_2$ with NaCl: development and application of a model for estimating
available surface areas and corrected uptake coefficients, Phys. Chem. Chem. Phys., 5, 1780-1789,
https://doi.org/10.1039/B301126G, 2003a.
Hoffman, R. C., Kaleuati, M. A., and Finlayson-Pitts, B. J.: Knudsen cell studies of the reaction of
gaseous $HNO_3$ with NaCl using less than a single layer of particles at 298 K: A modified mechanism, J.
Phys. Chem. A, 107, 7818-7826, https://doi.org/10.1021/jp030611o, 2003b.
Kerminen, V.-M., Teinilä, K., Hillamo, R., and Pakkanen, T.: Substitution of chloride in sea-salt
particles by inorganic and organic anions, J. Aerosol Sci., 29, 929-942,
https://doi.org/10.1016/S0021-8502(98)00002-0, 1998.
Ma, S. S., Yang, M., Pang, S. F., and Zhang, Y. H.: Subsecond measurement on deliquescence kinetics
of aerosol particles: Observation of partial dissolution and calculation of dissolution rates,
Chemosphere, 264, 128507, https://doi.org/10.1016/j.chemosphere.2020.128507, 2021.
Wang, X. W., Jing, B., Tan, F., Ma, J. B., Zhang, Y. H., and Ge, M. F.: Hygroscopic behavior and
chemical composition evolution of internally mixed aerosols composed of oxalic acid and ammonium
sulfate, Atmos. Chem. Phys., 17, 12797-12812, https://doi.org/10.5194/acp-17-12797-2017, 2017.

---

## Author Response (AR1)

**1 **Response to Reviewers:**

Thanks for the reviewer's comments on our manuscript entitled "A comprehensive study on hygroscopic behaviour and nitrate depletion of NaNO3 and dicarboxylic acid mixtures: Implication for the influence factors of nitrate depletion". The reviewers' comments are helpful for improving the quality of our work. The responses to the comments and the revisions in manuscript are given point-to-point below.

7

**8 **Reviewer #1:**

- In the abstract, the authors suggested that "The HNO3 release from NaNO3/OA mixtures was
   observed in both the measurements, owing to the relatively high acidity of OA". What does
   "the relatively high acidity of OA" mean? Compared with MA and GA, or HNO3? This
   should be revised to avoid misunderstanding.
- Author reply: Thanks for the reviewer's suggestion. The relatively high acidity of OA indicated that the acidity of OA was higher than MA and GA, but lower than HNO3. Indeed, this expression is a bit misleading. Thus, the sentence "The HNO3 release from NaNO3/OA mixtures was observed in both the measurements, owing to the relatively high acidity of OA" has been revised to "The HNO3 release from NaNO3/OA mixtures was observed in both the measurements, owing to the relatively higher acidity of OA compared to MA and GA" in the revised manuscript.
- 20 2. Line 37: Is there considerable amounts of nitrates present in sea salt aerosols? Or did the21 nitrate depletion frequently occur in sea salt aerosols?

Author reply: As known, nitrogen oxides in the atmosphere can undergo heterogeneous
 reactions with NaCl on the highly reactive surfaces of sea salt particles. The proposed reaction
 pathways are as follows (Gibson et al., 2006).

- 25  $\operatorname{NaCl}(s, aq) + \operatorname{HNO}_3(g) \rightarrow \operatorname{NaNO}_3(s, aq) + \operatorname{HCl}(g)$  (R1)
- 26  $\operatorname{NaCl}(s, aq) + \operatorname{N}_2O_5(g) \rightarrow \operatorname{NaNO}_3(s, aq) + \operatorname{ClNO}_2(g)$  (R2)
- 27  $\operatorname{NaCl}(s, aq) + 2\operatorname{NO}_2(g) \rightarrow \operatorname{NaNO}_3(s, aq) + \operatorname{ClNO}(g)$  (R3)
- Furthermore, the heterogeneous uptake coefficient ( $\gamma$ ) of HNO3 and N2O5 on NaCl particles has been measured to be about ( $1.0 \pm 0.8$ ) × 10-3 and ( $2.9 \pm 1.7$ ) × 10-3, respectively

(Hoffman et al., 2003a;Hoffman et al., 2003b). Thus, the chloride in sea salt aerosols can be
substituted significantly by nitrate, as observed in field measurements by Kerminen et al.
(1998). Coupled with the gas-particle partitioning of dicarboxylic acids, we can infer that
nitrate depletion by DCAs would occur frequently in sea salt aerosols.

34 3. The authors observed the  $NaHC_2O_4$  formation in  $NaNO_3/OA$  mixed system, and then 35  $NaHC_2O_4$  was transformed into  $Na_2C_2O_4$  with further nitrate depletion. Whereas, MA was 36 found to produce monosodium malonate as it reacted with nitrate. Why? Please clarify it in 37 the text.

**Author reply:** Thanks for the reviewer's suggestion. The OA has a  $pK_{a2}$  of 4.19, which is lower than that of MA ( $pK_{a2} = 5.70$ ). This implied that more  $C_2O_4^{-2}$  were dissociated from OA than  $CH_2(COO)_2^{-2}$  from MA. Thus, the disodium oxalate was mainly formed in NaNO3/OA system, while monosodium malonate was the main product as MA reacted with NaNO3. We have added the sentence "Besides, no disodium salts are observed in the NaNO3/MA system, differing from the NaNO3/OA system, which may be due to the higher acidity of OA ( $pK_{a2} =$ 4.19) than MA ( $pK_{a2} = 5.70$ )." in Line 270.

4. Line 132: "Likewise, Wang et al. (2017) observed the formation of NH4HC2O4 in mixed
(NH4)2SO4/OA droplets upon drying." So what's the similarity of Wang's study and this work?
Did they propose the similar driving force for HC2O4- ions formation?

**48 Author reply:** Wang et al. (2017) has indicated that OA could react with  $(NH_4)_2SO_4$  to form

(R5)

- 49  $NH_4HSO_4$  and  $NH_4HC_2O_4$  via the following pathway:
- 50  $H_2C_2O_4(aq) \to H^+(aq) + HC_2O_4^-(aq)$  (R4)

51
$$H^+(aq) + SO_4^{2-}(aq) \rightarrow HSO_4^-(aq)$$

52
$$2NH_4^+(aq) + HSO_4^-(aq) + HC_2O_4^-(aq) \rightarrow NH_4HSO_4^-(aq) + NH_4HC_2O_4^-(aq)$$
 (R6)

In the first step, the OA ( $pK_{a1} = 1.23$ ) was dissociated into HC2O4- and H+ ions. Then, the H+ reacted with SO42- to form HSO4- ions. While in this work, the dissociated H+ ions would interact with NO3- to produce gaseous HNO3, causing nitrate depletion. After that, the HC2O4- ions further combined with NH4+ to produce NH4HC2O4 in Wang's work, and in this study, the HC2O4- ions tended to combine with Na+ to generate NaHC2O4. Besides, the formation of both NH4HC2O4 and NaHC2O4 was observed in the drying process in the two studies. 59 Therefore, We proposed that "Likewise, Wang et al. (2017) observed the formation of 60  $NH_4HC_2O_4$  in mixed ( $NH_4$ )2SO4/OA droplets upon drying.".

- 5. Line 143: As indicated in this work, NaNO3 deliquescence proceeds at 46.9%-61.9% RH for
  NaNO3/OA mixtures and ~65%-77% RH for pure NaNO3. This implied that NaNO3 solids
  began to deliquesce at RH significantly lower than its DRH. So is the deliquescence process a
  thermodynamic process or a kinetic process?
- Author reply: In our previous work, the NaNO3 solids were found to begin to dissolve at  $\sim 63.70\%$  RH during the humidification, which was well below the predicted DRH by EAIM model (Ma et al., 2021). Meanwhile, the deliquescence behavior of inorganic salt particles (i.e., NaCl, NaNO3 and K2CO3) has been proved to be a dynamic process, i.e., the particles would absorb water to form the partially dissolved phase state at RH lower than their DRH, and then they were deliquesced completely once the DRH was reached (Bruzewicz et al., 2011;Esat et al., 2018;Ma et al., 2021).
- Line 168: The authors observed phase transition and nitrate depletion of 1:1 NaNO3/OA
  mixtures in Sec. 3.1, but why did they choose 3:1 mixtures to further investigate the phase
  state effect?
- Author reply: As indicated in this work, the final reaction product of NaNO3/OA system was Na2C2O4. Thus, we chose 3:1 NaNO3/OA mixtures, in which the amount of reactant NaNO3 was in excess according to stoichiometry, to examine whether the liquid OA could be consumed completely by aqueous NaNO3. Based on this, we indicated that "At ~ 69.2% RH, the presence of 1741 cm-1 band indicates the excess of liquid OA, suggesting this displacement reaction tends to reach equilibrium with comparable final concentrations of "reactants" and "products".".
- Line 231: The 15% RH was only a preset RH value with stepwise increasing RH, so it should
  be revised to "As RH increases to around 15% (or even lower)".
- Author reply: Thanks for the reviewer's suggestion. We have adopted reviewer's advice and
   revised our manuscript accordingly.
- 86 8. Line 287: The authors also observed the chlorine depletion in 1:1 mixed NaCl/MA particles
  87 with two different RH changing rates, but a brief discussion about the experimental results

88 should be presented in the text.

Author reply: Thanks for the reviewer's suggestion. We have adopted reviewer's advice andrevised our manuscript accordingly.

9. Line 413-424: I suggest that this paragraph should be rewrote to better illustrate the
atmospheric implications of the experimental observations in view of the presence of mineral
dust inclusions and so on in atmospheric aerosols, which constantly induce the heterogeneous
nucleation of aerosols at relatively high RH.

- 95 Author reply: Thanks for the reviewer's suggestion. We have adopted reviewer's advice and 96 rewrote this paragraph to illustrate the atmospheric implications of the influence of 97 efflorescence behaviors of aerosols on nitrate depletion.
- 98

**99 Reviewer #2:**

**100 Major comments**

101 1. My main concern is with the presentation and interpretation of several of the results with 102 regard to the loss of nitrate from the particles/droplets via evaporation of HNO3. As the title 103 indicates, nitrate depletion is a key interest of this work, yet most of the data provided does 104 not appear to show direct evidence for nitrate depletion via HNO3 loss to the gas phase -105 contrary to what is stated in the abstract (line 15) and the Results and Discussion section. The 106 measurements provided characterize the composition and spectral features of nitrate and DCA 107 species in the condensed phase(s) of deposited particles. However, unless I missed it, HNO3(g) 108 is not measured, nor is the mass of nitrate in the particle phase tracked. If there is substantial 109 loss of nitrate, the particles should change in nitrate content and overall mass. Moreover, the 110 loss of nitrate would likely be accompanied by a loss of water. As pointed out in the specific 111 comments that follow, there are several instances where statements like "... confirms nitrate 112 depletion and HNO3 release" are made (e.g. line 133), while the information provided does 113 not directly support this conclusion. For example, the authors seem to equate the formation of 114 liquid or solid NaHC2O4 as evidence for nitrate depletion. However, it is not discussed why 115 that should be a unique signature of such a process. This reviewer argues that (at least some) 116 dissolved or crystalline NaHC2O4 could also form in the absence of any nitrate depletion,

simply as a result of aqueous solution chemistry and resulting equilibria. I strongly suggest that the authors rectify this issue and provide direct evidence for HNO3 loss to the gas phase or otherwise clarify that their interpretation depends on certain assumptions about the relevant processes taking place.

121 To be clear, I also consider nitrate depletion via HNO3 to be a plausible hypothesis and a 122 likely explanation for the findings from this work. However, what is missing is quantitative 123 evidence and/or clarity in the discussion in support of this hypothesis.

124 Author reply: First of all, the HCl/HNO3 release from mixed nitrate/chloride and organic acid particles has been widely detected in field and laboratory measurements (Laskin et al., 125 2012; Wang and Laskin, 2014; Ma et al., 2019a; Ghorai et al., 2014; Shao et al., 2018). Laskin's 126 group has employed the computer controlled scanning electron microscopy with energy 127 128 dispersed analysis of X-rays (CCSEM/EDX) to measure the Cl/Na ratios in mixed organic 129 acid/NaCl particles (Laskin et al., 2012;Ghorai et al., 2014) and N/Na ratios in mixed organic acid/NaNO3 particles (Wang and Laskin, 2014). The measurement results clearly 130 131 demonstrated the significant chloride/nitrate depletion by organic acids. Ma et al. (2013) 132 measured the water adsorption isotherms of mixed NaCl/OA particles using vapor sorption analyzer. The decreased particle mass, i.e., water content therein, indicated the formation of 133 134 less hygroscopic Na2C2O4 in internally mixed particles. The corresponding Raman spectra 135 results also confirmed the chloride depletion. Li et al. (2017) measured the Raman features of 136 mixed NaCl/MA particles using in situ Raman microspectrometry and found the production of 137 monosodium malonate in mixtures during the RH cycle, which demonstrated the chloride 138 depletion and gaseous HCl release. Furthermore, the specific reactions of strong acids 139 displaced by weak acids in aerosols have been summarized and discussed in our preivous 140 review (Chen et al., 2021). Second, our group has persistently utilized the FTIR spectra to 141 characterize the chemical composition evolutions of aerosol reaction systems (Wang et al., 142 2019;Shao et al., 2018;He et al., 2017;Du et al., 2020). For instance, Wang et al. (2019) 143 observed the ammonium depletion and gaseous NH3 release from mixed dicarboxylic acid 144 salts and  $(NH_4)_2SO_4$  aerosols during the RH cycle using the ATR-FTIR technique. Similar to 145 this work, they measured the IR feature changes of dicarboxylic acid salts and corresponding

146 DCAs, as well as the efflorescence transition behaviors of mixed particles. The results clearly 147 demonstrated the compositional evolution and ammonium depletion in mixed dicarboxylic acid salts and (NH4)2SO4 aerosols. Shao et al. (2018) determined the chemical composition 148 149 evolution of MA/NaNO3, MA/Mg(NO3)2 and MA/Ca(NO3)2 particles by vacuum FTIR method. The intensity changes of feature bands of COO-, COOH and NO3- groups indicated 150 the formation of malonate salts and HNO3 release. Finally, the almost unchanged water 151 content after nitrate depletion for DCA/NaNO3 mixtures in this work, e.g., MA/NaNO3 152 153 mixtures in Fig. 5a in the revised manuscript, might be due to the comparable hygroscopic 154 properties of malonate salts and MA (Jing et al., 2018; Wu et al., 2011), or limited amounts of 155 formed dicarboxylic acid salts.

156 Specific comments

- Title: phrasing of "Implication for the influence factors of nitrate depletion" needs language
   improvement and perhaps a more direct link to aerosols. Consider replacing by "implications
   for nitrate depletion in tropospheric aerosols".
- Author reply: Thanks for the reviewer's suggestion. We have adopted reviewer's advice and
  revised our manuscript accordingly.
- 162 3. Line 15: make statement consistent with the data provided (see my major comment).

Author reply: Thanks for the reviewer's suggestion. We have illuminated the agreements on the observations of nitrate/chlorine depletion and HNO3/HCl release from mixed nitrate/chlorine and DCA particles via various measurement techniques (including FTIR spectra) from previous literatures. For clarity, we have revised "HNO3 release" into "nitrate depletion" in Line 15.

- Line 19: "... potentially indicating the transformation of amorphous solids to semisolid
  NaNO3; ..."; consider: or the dissolution of NaNO3 into a liquid DCA + Water rich phase?
- 170 Author reply: As shown in Fig. S6, no obvious changes of the 1356 cm-1 band assigned to
- 171  $v_3(NO_3^{-})$  of amorphous NaNO3 solids were observed at constant ~ 15% RH. Meanwhile, the
- 172 liquid water content remained almost unchanged as RH increased from  $\sim 14.8\%$  to  $\sim 61.0\%$
- 173 (seen Fig. 3b). Therefore, we can conclude the transformation of amorphous solids to
- 174 semisolid NaNO3, rather than the dissolution of NaNO3 solids as RH increased to  $\sim 15\%$ .

175 5. Line 39: define abbreviations like DCA, ATR-FTIR and OA at first use in the main text (aside176 from the abstract).

Author reply: Thanks for the reviewer's suggestion. We have adopted reviewer's advice and
revised our manuscript accordingly.

- 179 6. Line 42: there are several directly relevant studies on hygroscopicity and surface tension
  180 effects that could be cited here; e.g. Peng et al. (2001, 10.1021/es0107531), Ovadnevaite et al.
- 181 (2017, doi:10.1038/nature22806), Hodas et al. (2015, doi:10.5194/acp-15-5027-2015).
- 182 Author reply: Thanks for the reviewer's suggestion. We have adopted the reviewer's advice183 and revised our manuscript accordingly.
- 7. Line 43: "It is well known that the displacement of strong acids, i.e., HCl or HNO3, by weak 184 organic acids, e.g., DCAs, is not thermodynamically favoured in bulk solution". This 185 186 statement requires citation of a relevant reference. Note that even if the thermodynamic 187 equilibrium is favouring the left hand side of reaction (R1), there will be some amounts of 188 HCl or HNO3 dissolved and if gas phase exchange is possible, there will be a continuous loss 189 of the right hand side, consuming the reactants of R1 over possibly long time (depending on 190 bulk vs. small aerosols). Therefore, given that in an aqueous phase, NaNO3 will be present 191 mostly in the form of  $Na^+$  and  $NO_3^-$ , some NaA(aq) (or solid) and/or  $Na^+$  and  $A^-$  may be 192 present even in the absence of any evaporation of HNO3.
- Author reply: Thanks for the reviewer's suggestion. The HNO3 and HCl are the stronger 193 194 acids compared to DCAs, and the displacement of weak DCAs by strong HNO3 or HCl is 195 thermodynamically favoured in bulk solutions. Contrary to this, the displacement of strong 196 acids by weak organic acids is not thermodynamically favoured in bulk solutions. While in 197 aerosol phases, the equilibrium of reaction R1 would shift to the right due to the efficient 198 evaporation of HCl from the particle phase to the gas phase, in view of the much higher 199 surface-to-volume ratios of aerosols compared to bulk solutions. Indeed, in an aqueous phase, NaNO3 would be present mostly in the form of Na+ and NO3-, and HA can be dissociated into 200 201  $H^+$  and  $A^-$ . Whereas, not considering the displacement reaction equilibrium, the amounts of 202 dissociated  $H^+$  and  $A^-$  are limited according to the acid dissociation constants of organic acids. 203 Based on this, the formed NaA (aqueous or solid) and dissolved HCl or HNO3 are very

- 204 limited and cannot be detected by our IR spectra (seen Fig. 5d and 6d). Furthermore, HCl and 205 HNO3 have the very high volatility, i.e., low Henry's law constants (HCl:  $< 2 \times 10^{-1}$ ; HNO3: > 206  $2 \times 10^{5}$ ), and thus they tend to partition from the particle phase to the gas phase, causing the 207 shift of the reaction R1 to the right and much more NaA formation.
- 208 8. Line 52: State for what concentration/conditions. Are you comparing at the same RH or the
  209 same water content or the same molar concentration of dissolved solutes?
- 210Author reply: Thanks for the reviewer's suggestion. We have stated that in mixed211NaCl/citric acid droplets with molar ratio of 1:1, dissociated HCl concentration is ~  $10^{10}$ 212times higher than dissociated citric acid according to the different acid dissociation constants213 $(K_{a1})$  (HCl: > 1 × 107, citric acid:  $8.4 \times 10^{-4}$ ) (Laskin et al., 2012).
- 214 9. Line 58: define the base of the logarithm and perhaps at first use, define the meaning of pKa1
  215 formally using molal activities or standard chemical potentials, as appropriate given the
  216 values stated.
- Author reply: Thanks for the reviewer's suggestion. We have adopted the reviewer's adviceand revised our manuscript accordingly.
- 219 10. Line 66: what "substrate" is here referred to? do you mean material onto which particles were220 deposited in laboratory experiments?
- 221 Author reply: The substrate materials referred to the substrates on which particles were 222 deposited in the previous studies. For instance, in Ghorai's study, the submicrometer 223 NaCl/MA(GA) particles were deposited on  $Si_3N_4$  windows and TEM grids (Ghorai et al., 224 2014). In another study, the micron-sized NaNO3/MA(GA) particles were deposited on 225 carbon-filmed grids or  $Si_3N_4$  windows (Wang and Laskin, 2014). While Ma et al. (2013)
- 226 placed mixed NaCl/MA(GA) particles in aluminum sample holder or on Ge substrate.
- Line 70: "... the substrate supporting OA droplets would crystallize to form"; do you mean the
  droplet / OA or really the substrate? Clarify phrasing.
- Author reply: Thanks for the reviewer's suggestion. We have revised "substrate supporting
  OA droplets" into "OA droplets deposited on polytetrafluoroethylene (PTFE) substrate".
- 231 12. Line 70: "OA dihydrate at 71% RH" Is it under conditions of dehumidification? Describe
  232 what kind of laboratory experiment you refer to.

Author reply: Thanks for the reviewer's suggestion. We have revised "at ~ 71% RH" into "at
 ~ 71% relative humidity (RH) during dehydration".

- Line 72: "Pure NaNO3 droplets might not be effloresced..." By "be effloresced" do you rather
  mean "crystallize" here? Note that effloresced is not necessarily the same as crystalline in the
  context of amorphous solids as a possibility. Also, clarify under what conditions; here
  presumably dehydration in an isothermal (and isobaric) experiment.
- Author reply: Thanks for the reviewer's suggestion. "Pure NaNO3 droplets might not be
- effloresced but convert into amorphous state at low RH" indicated that NaNO3 droplets could
- 241 neither be crystallized, nor be effloresced to amorphous solids, but instead formed viscous
- supersaturated liquids at low RH (Liu et al., 2008). For clarity, we have revised "amorphous
- state at low RH" into "highly concentrated droplets at low RH upon drying" in the text.
- 14. Line 73: "certain RH values" Which ones? Please provide values/ranges.
- Author reply: Thanks for the reviewer's suggestion. We have adopted the reviewer's adviceand revised our manuscript accordingly.
- Line 74: Given the stated list of options, how is "highly viscous" different from glassy or
  semisolid? One could say "highly viscous (i.e. semisolid or glassy), liquid, or mixed-phase
  (e.g. solid–liquid) mixing states.
- Author reply: Thanks for the reviewer's suggestion. We have adopted the reviewer's adviceand revised our manuscript accordingly.
- Line 75: Koop et al. (2011, doi:10.1039/C1CP22617G) would be another key reference to cite
  here.
- Author reply: Thanks for the reviewer's suggestion. We have adopted the reviewer's adviceand revised our manuscript accordingly.
- 256 17. Line 82: phrasing: "would" or rather "does" it?
- **Author reply:** Thanks for the reviewer's suggestion. We have adopted the reviewer's advice
- and revised our manuscript accordingly.
- 259 18. Line 85: delete "mixture"; it is redundant with solution.
- 260 Author reply: Thanks for the reviewer's suggestion. We have adopted the reviewer's advice
- and revised our manuscript accordingly.

262 19. Line 92: "... a brief was introduced here"; phrasing.

- Author reply: Thanks for the reviewer's suggestion. We have adopted the reviewer's adviceand revised our manuscript accordingly.
- 20. Line 103: "chamber, which could be used to calibrate the ambient RH"; in the context of this
  description, it is mentioned that water vapor and CO2 were removed by the vacuum pump. So,
  how was the RH established in the sample chamber? Is it due to remaining water vapor under
  low pressure conditions? If so, does that mean that the sample droplets are (potentially)
  evaporating water alongside with HNO3 and DCA? The droplets may also outgas dissolved
  CO2 during the experiment given the vacuum applied, which might have some effect on the
  acidity.
- 272 Author reply: Before the measurements, the water vapor and  $CO_2$  were removed by the 273 vacuum pump and the pressure in the sample chamber could arrive to  $\sim 0.01$  kPa. After that, 274 water vapor from the water reservoir was fed into the sample chamber to establish the ambient 275 RH in the sample chamber. When the outlet of water vapour was closed, the water vapour 276 pressure in the sample chamber would keep equilibrium with that of water reservoir, and the 277 maximum RH at the certain temperatures could be reached. In addition, the liquid water in the sample droplets did evaporate alongside with very small amounts of HNO3 and DCAs in the 278 279 vacuuming process. The HNO3 release and DCA salts formation during vacuuming have been indicated in the text. Besides, the dissolved  $CO_2$  in micron-sized droplets was very limited, 280 281 and it was conceivable that the influence of outgassing of dissolved CO2 on droplet acidity, 282 and thus the nitrate depletion in mixed NaNO3/DCA particles, was negligible.
- 283 21. Line 127: "and anhydrous OA" I suggest to mention that you refer to crystalline anhydrous284 OA here.

**Author reply: Thanks for the reviewer's suggestion. We have adopted the reviewer's adviceand revised our manuscript accordingly.**

287 22. Line 131: "... indicative of the NaHC2O4..."; Is this referring to an actual solid NaHC2O4 288 component forming or rather the presence of partially dissociated OA anions (HC2O4-) in 289 aqueous solution alongside Na+? In the latter case, this should not be referred to as NaHC2O4 290 since it is not forming a compound of that structure nor stoichiometry. Also, in that case, this does not confirm directly the release of HNO3, it only indicated the presence of partially
dissociated OA. Please discuss.

Author reply: Thanks for the reviewer's suggestion. We have revised "NaHC2O4" into "dissociated  $HC_2O_4$ " in the revised manuscript. As indicated above, the amounts of dissociated  $HC_2O_4$ " from OA were very limited and could not be detected by our IR spectra assuming that no displacement reactions occurred. Furthermore, the release of HNO3, which has been demonstrated in the earlier studies and discussed above, would shift the reaction R1 to the right. Therefore, more OA was dissociated into  $HC_2O_4$ " in the presence of Na+ along with continuous HNO3 release, as indicated in reaction R2.

300 23. Line 132: "Likewise, Wang et al. (2017) observed the formation of  $NH_4HC_2O_4$  in mixed 301  $(NH_4)_2SO_4/OA$  droplets upon drying. These scenarios confirm the nitrate depletion and  $HNO_3$ 302 release from NaNO3/OA mixtures in the vacuuming process." The provided data and 303 discussion does not uniquely support this conclusion. While  $HNO_3$  evaporation is possible, 304 where is the evidence for it? Oxalate salts could form also without any loss of  $HNO_3$  to the 305 gas phase. For example,  $HNO_3(aq)$  could form or, more likely,  $H^+(aq)$  and  $NO_3^-$  (aq) 306 alongside solid or dissolved OA salts. Please provide evidence or a more nuanced discussion.

**Author reply:** As discussed in the author reply for the major comment, the HNO3 release from mixed nitrate and organic acid particles has been widely detected in field and laboratory measurements. Also, the HNO3 in the aqueous phase tends to partition into the gas phase due to the high volatility, i.e., low Henry's law constants (>  $2 \times 10^5$ ), of HNO3 and the high surface-to-volume ratios of aqueous droplets. Therefore, the formation of OA salts would be accompanied by HNO3 release, in other words, the oxalate salts formation did confirm the nitrate depletion and HNO3 release from NaNO3/OA mixtures.

24. Line 145: "Likewise, Ma et al. (2013) found that the DRH of NaCl component decreased in
both external and internal NaCl/MA mixtures." How could this be the case in an external
mixture (assuming external means (pure) NaCl particles separate from MA particles)?

Also, worth mentioning in the context of DRH lowering: a lowering of the DRH in particles
 containing aqueous DCAs is expected from thermodynamic equilibrium theory and has been

319 shown in many other measurements and by means of thermodynamic model predictions, such

320 as those shown for MA or OA and ammonium sulfate by Bouzidi et al. (2020, doi:10.1016/j.atmosenv.2020.117481) and those shown by Hodas et al. (2015, 321 322 doi:10.5194/acp-15-5027-2015) for mixtures of ammonium sulfate with DCAs and citric acid Author reply: Thanks for the reviewer's suggestion. In Ma's study, they found that externally 323 and internally mixed NaCl/MA particles exhibited DRH in the range of 65-70% RH and 60-65% 324 325 RH, respectively, which were lower than the DRH of pure NaCl particles (75% RH). Indeed, it might be implausible that the dissolution of external NaCl particles were influenced by 326 327 separated MA particles. One possible reason for this might be that the contact of solution films of separated NaCl and MA particles, caused by water absorption of solid particles at RH 328 329 below the DRH (Bruzewicz et al., 2011; Wise et al., 2008), led to the decrease in surface 330 tensions of solutions and enhancement of water absorption, which further reduced the DRH of 331 pure components. Whereas for clarity, we have deleted this sentence and added the citations 332 of relevant studies by Bouzidi et al. and Hodas et al.

333 25. Line 146: "Note that the OA dihydrate and crystalline Na2C2O4 cannot be deliquesced due to 334 their very high DRHs ...". A note: the process of deliquescence, as more clearly observed in 335 binary (water + 1 solute) systems, should not be confused with partial (gradual) dissolution in 336 multicomponent, multiphase systems. Therefore, please clarify the sentence. At present it is 337 misleading. While crystalline substances like OA and Na2C2O4 may not fully dissolve, it 338 should be noted that in the presence of an aqueous phase a solid-liquid equilibrium will be 339 established (given enough time) and some amount of OA and Na2C2O4 will be in the 340 dissolved state, forming a saturated aqueous solution with respect to the pertaining crystalline 341 phase phases; the discussion Hodas al. (2016, or see. e.g. bv et 342 doi:10.5194/acp-16-12767-2016).

343 Author reply: Thanks for the reviewer's suggestion. We have adopted the reviewer's advice344 and revised our manuscript accordingly.

26. Line 155: "Upon hydration, the water content at high RH is far below that upon
dehydration, …" This statement needs further clarification. It seems only to make sense when
the dehydration refers to dehydration from an initial state of completely dissolved particles at
very high RH. Of course, one could start dehydration at any RH and then the water content

- could be higher or lower than when arrived at that state from an initial state of low RH.
- Author reply: Thanks for the reviewer's suggestion. We have adopted the reviewer's adviceand revised our manuscript accordingly.
- 27. Line 159: I find Fig. S6 insightful and suggest that this figure and the related discussion bemoved to the main manuscript.
- Author reply: Thanks for the reviewer's suggestion. We have adopted the reviewer's adviceand revised our manuscript accordingly.
- 28. Line 160: "Besides, as shown in Fig. 1c, the absorbance of 1620 cm-1 band assigned to
  oxalate shows a slight increase at RH as low as 21.2%, implying the nitrate depletion
  proceeds at relatively low RH."
- In my opinion, what is missing are observations of the loss of nitrate/HNO3 from the particles. Do the authors have comparison examples for cases where the gas phase exchange is limited (small gas volume) such that only a small loss could occur and, as a result, the signature in oxalate formation substantially different? Without such observations, or other quantitative measures of loss to the gas phase, the interpretation of the experiments hinges on assuming a nitrate depletion takes place without having the data to confirm it. Please discuss.
- Author reply: As already indicated, Li et al. (2017) observed the Raman features of 365 366 monosodium malonate in NaCl/MA mixed aerosols during the RH cycle, indicating the chloride depletion and gaseous HCl release. Wang et al. (2019) confirmed the ammonium 367 depletion and gaseous NH3 release from mixed dicarboxylic acid salts and (NH4)2SO4 368 aerosols during the RH cycle according to the IR feature changes of dicarboxylic acid salts 369 370 and corresponding DCAs via the ATR-FTIR technique. Ma et al. (2013) also determined the 371 oxalate formation and gaseous HCl release from NaCl/OA mixtures based on the Raman and 372 ATR-FTIR characterization results. Therefore, the increase in absorbance of IR feature bands 373 of oxalate is proved to be powerful evidence for nitrate depletion and gaseous HNO3 release.
- 29. Line 173: "As RH increases to ~ 15%, the  $\Delta A$  of  $v_{as}(COO-)$  band exhibits a considerable increase, indicative of the occurrence of nitrate depletion."

While indicative as an option, it remains speculative. It is indicative of the partial dissociationof OA in solution. Which is expected to increase with increasing particle water content and,

- hence, increasing RH.
- **Author reply:** As already indicated, there were no IR feature bands of COO-, e.g., 1465 cm-1
- band, in IR spectra of pure OA droplets, as shown in Fig. S3. Thus, the increase in  $\Delta A$  of  $v_{as}(COO^{-})$  band did indicate the occurrence of nitrate depletion.
- 382 30. Line 177: "2013). Then, the 1620 cm-1 band appears and becomes stronger with time, 383 suggesting Na2C2O4 can be continuously produced at constant ~ 15% RH."
- 384 I think this is better evidence for Na2C2O4 formation and associated nitrate depletion, because
- 385 the system is observed at constant RH, which means it should maintain constant water activity
- 386 (in sufficiently large particles where the Kelvin effect would not change with evaporation). It
- 387 may be good to expand the discussion of this piece of evidence.
- 388 Author reply: Thanks for the reviewer's suggestion. We have adopted the reviewer's advice389 and revised our manuscript accordingly.
- 390 31. Line 179: "... we can infer the conversion of amorphous NaNO3 solids to highly viscous
  391 semisolids due to the uptake of trace amounts of moisture."
- Please clarify: semisolids of what? if NaNO3 dissolves partially and forms a viscous aqueous
  solution, this should not be called a NaNO3 semisolid because the formed phase contains
  other species too.
- 395 Author reply: Thanks for the reviewer's suggestion. We inferred that amorphous NaNO3
- 396 solids did not dissolve partially, based on the IR feature changes of NO3- as discussed above,
- 397 but transformed into highly viscous NaNO3 semisolids. We have adopted the reviewer's
- 398 advice and revised our manuscript accordingly.
- 399 32. Line 216: Define the  $R^2$  metric.
- 400 Author reply: Thanks for the reviewer's suggestion. We have adopted the reviewer's advice401 and revised our manuscript accordingly.
- 402 33. Line 231: "As RH increases to around 15%, amorphous solids are converted into viscous
  403 semisolids, ...". Is there any evidence from the spectra for such a conversion and the
  404 approximate viscosity of the phase to classify it as semisolid?
- 405 Author reply: Thanks for the reviewer's suggestion. There were no spectra and viscosity data
- 406 for the conversion of amorphous solids to semisolids. First of all, NaNO3 has been proved to

407 exist in unusual metastable states, e.g., amorphous solids or highly concentrated droplets, after

- 408 drying (Ma et al., 2021;Liu et al., 2008;Hoffman et al., 2004). Then, the chemical reactivity to
- 409 liquid OA did alter as RH increased to  $\sim 15\%$  herein. Coupled with the particle morphology
- 410 changes shown in Fig. 2, we can speculate that amorphous NaNO3 solids were converted into
- 411 viscous NaNO3 semisolids as RH increased to  $\sim 15\%$ .
- 412 34. Line 261: "... formation of monosodium malonate ...". Is this for a solid malonate phase or413 dissolved?
- 414 Author reply: Li et al. (2017) measured the Raman spectra of pure monosodium malonate (MSM) aerosols at RH = 10% and 2:1 NaCl/MA mixed aerosols at very low RH of 9.1% upon 415 416 dehydration, which did not resemble that of MSM powders, suggesting that the pure MSM aerosols and mixed NaCl/MA aerosols tended to form amorphous states after drying. 417 Furthermore, in this work, the IR feature band of MSM, i.e., 1595 cm-1 band, showed no 418 obvious changes during humidification, as shown in Fig. 5c, indicating no deliquescence 419 420 transition of MSM occurrence. Based on these, we can infer that the MSM component in 421 mixed NaNO3/MA aerosols was in amorphous liquids, rather than solid phases.
- 422 35. Line 266: Clarify: "that the first acid dissociation constant (pKa1) of MA was about 2.83,
  423 which was ~ 3 orders of magnitude larger than the second one ...". 3 orders of magnitude in
  424 what? clearly not in pKa. Also, 2.83 is the pKa1 not Ka1 (which the current phrasing implies).

**425 Author reply: Thanks for the reviewer's suggestion. We have adopted the reviewer's advice426 and revised our manuscript accordingly.**

- 427 36. Line 270: band assigned to  $v_{as}(COO^{-})$  of monosodium malonate. Please clarify: why of 428 monosodium malonate and not of the malonate anion? The  $COO^{-}$  functionality does not 429 contain any sodium and in aqueous solution the malonate ion is likely dissociated from Na+. Is
- 430 the assigned band specific to the COO- group in crystalline monosodium malonate only?
- 431 **Author reply:** Thanks for the reviewer's suggestion. We have adopted the reviewer's advice
- 432 and revised our manuscript accordingly.
- 433 37. Line 277: "aqueous NaNO3" should be "aqueous NO3.
- 434 Author reply: Thanks for the reviewer's suggestion. We have adopted the reviewer's advice
- 435 and revised our manuscript accordingly.

436 38. Line 279: "As compared to vacuum FTIR results, we can infer that the heterogeneous efficacy
437 of Ge substrate is much higher than CaF2 windows,...". It seems unclear whether this is really
438 a substrate effect or rather an effect of the vacuum cell procedures applied

Author reply: In our previous review, the efflorescence kinetics and nucleation mechanisms
were discussed detailed. The substrate effects have been proved to be a key factor to induce
heterogeneous nucleation of aerosol particles. Furthermore, the heterogeneous nucleation
efficiency of CaF2, Ge and ZnSe substrates for deposited particles have been demonstrated in
our earlier studies (Ma et al., 2019b;Ji et al., 2017;Zhang et al., 2014;Ren et al., 2016).

- Line 290, Fig. 5: What explains the difference in the normalized water contents comparing
  panels (a) and (b)? Did you compare the obtained water contents to model predictions of the
  water content or independent mass growth factor measurements to assess which technique
  provides more accurate hygroscopic cycles?
- 448 Author reply: The difference in the normalized water content in panels (a) and (b) was 449 attributed to the different sequences of water cycles. In other words, the deposited droplets in 450 **FTIR** first vacuum were dried in vacuum, and then underwent a 451 humidification-dehumidification cycle. According to the weak absorbance intensity of 1548 cm-1 band shown in Fig. 6c in the revised manuscript, we can infer that the nitrate depletion 452 453 was limited and might not alter the hygroscopic ability of mixed aerosols. While for 454 ATR-FTIR measurement, the mixed droplets first underwent the dehumidification process and 455 then a humidification process. Upon hydration, the GA component mainly existed in solid 456 state and could not be dissolved completely even until 84% RH, and thus the water content 457 could not match with that in the dehumidification process. Note that the complete 458 deliquescence of mixed NaNO3/GA particles in vacuum FTIR measurement might be due to 459 the partial crystallization of GA in the vacuuming process.
- Firstly, the formation and precipitation of organic acid salts have not been considered by the current version of E-AIM, thus, the model cannot provide accurate hygroscopic growth factors for NaNO3/DCA mixtures. Second, the mixed aerosols (e.g., NaNO3/OA mixtures) might exist in solid-liquid mixing state, not the aqueous solution state at high RH in the vacuum FTIR measurement. Therefore, we could not determine the mass growth factors of

465 mixed particles through the absorbance of liquid water band and E-AIM predictions (even 466 though the data was not accurate), according to the calculation method proposed by our earlier 467 studies (Ma et al., 2019b;Ji et al., 2017). Note that the mass growth factors and E-AIM 468 predictions of pure NaNO3 and DCA aerosols have been determined and shown in the 469 Supplement.

- 470 40. Line 301: "appears upon hydration, indicating the formation of glutarate sodium salts." Could
  471 you clarify whether this means that a solid crystalline salt phase is forming upon hydration in
  472 equilibrium with an aqueous solution?
- 473 Author reply: Thanks for the reviewer's suggestion. According to the hygroscopic 474 behaviours of mixed NaNO3/GA particles and IR feature changes of  $v_{as}(COO^{-})$ , which was 475 similar to that of mixed NaNO3/MA particles, we can infer that the formed glutarate sodium 476 salts were present in aqueous solution state. Furthermore, we have revised our manuscript 477 accordingly.
- 41. Line 314: "lower temperature of droplets" Is it just the temperature effect or also the drying tomuch lower effective RH than during the ATR-FTIR measurement cycles?
- 480 Author reply: The lowest RH in the two measurements was comparable, i.e., 3.4% RH in 481 vacuum FTIR vs. 5.6% RH in ATR-FTIR. The temperature effect caused by rapid water 482 evaporation has been introduced in our previous study, which can facilitate the nucleation of 483 aqueous droplets (Ma et al., 2019b).
- 484 42. Line 335: Could you discuss in this context whether the size dependence effect is due to the
  485 Kelvin effect, leading to a lower water activity and higher pH in smaller particles compared to
  486 larger ones exposed to the same RH in the gas phase, or some other effect?
- 487 Author reply: Thanks for the reviewer's suggestion. As indicated in our previous review, the 488 depletion extent,  $\xi$ , was related to depleted mass ( $\Delta m$ ) and initial mass ( $m_0$ ) of a volatile 489 species within a droplet with the radius r, which could be expressed as  $\xi = \Delta m/m_0$  (Chen et al., 490 2021). The  $\Delta m$  could be calculated by  $\Delta m = 4\pi r M D p_r/(RT\Delta t)$  via the Maxwell steady-state 491 diffusive mass transfer equation, where M, D, and  $p_r$  denoted the volatile species molecular 492 weight, diffusion coefficient in the air, and equilibrium partial pressure at the droplet surface, 493 respectively; R, T, and  $\Delta t$  were the ideal gas constant, temperature, and reaction time,

494 respectively. The  $m_0$  of the volatile species could be expressed as  $m_0 = 4\pi r^3 c/3$ , since the 495 concentration of the volatile species (c) within the droplet kept constant at a fixed RH. Based 496 on these, the  $\xi$  could be expressed as

497
$$\xi = \frac{k}{r^2} \Delta t, \text{ with } k = \frac{3MDp_r}{4\pi cRT}$$
(1)

498 As seen, the depletion extent,  $\xi$ , was inversely proportional to  $r^2$ . Therefore, the size 499 dependence effect of depletion extent was mainly due to the volatility difference, rather than 500 the Kelvin effect.

501 43. Line 346, Eq. (5): check equation; it looks like the p of  $p_r$  is missing.

502 Author reply: Thanks for the reviewer's suggestion. We have adopted the reviewer's advice503 and revised our manuscript accordingly.

44. Line 358: Eq. (6), for consistency with Eq. (7), consider replacing P by p\*.

505 Author reply: Thanks for the reviewer's suggestion. Firstly, we indicated the expression of 506 the diffusion coefficient of HNO3 in the gas phase according to Chapman-Enskog method. In 507 other words, the D denoted to the diffusion coefficient of HNO3 in the ATR-FTIR 508 measurement. Then, the HNO3 diffusion coefficient in vacuum FTIR measurement, which 509 was defined as  $D^*$ , was related to the ambient pressure in vacuum FTIR,  $P^*$ . Thus, the P in 510 the eq. (6) did not only denote to  $P^*$  in the eq. (7).

45. Line 373: "Therefore, the ambient pressure, dominating the diffusion coefficient of HNO3 in
the gas phase, tends to play an important role in nitrate depletion during the transport and
aging of atmospheric aerosols."

This statement is not supported by any quantitative data. How big of a difference is expected in the troposphere? Obviously, one should not directly compare the vaccuum FTIR pressure to realistic tropospheric conditions without accounting for the differences in the ranges of total pressures. If nitrate-containing particles are given hours to days to undergo displacement reactions, the impact of pressure on HNO3 diffusion may be unimportant. Do you have predictions to the contrary?

520 Author reply: Thanks for the reviewer's constructive suggestion. We have deleted the 521 improper statement, i.e., "Therefore, the ambient pressure, dominating the diffusion 522 coefficient of HNO3 in the gas phase, tends to play an important role in nitrate depletion

- 523 during the transport and aging of atmospheric aerosols." in the revised manuscript.
- 46. Line 409: "NaNO3 can be treated as a surrogate for a broad class of amorphous or semisolid
  species existing in atmospheric aerosols,..."
- 526 Please state in what RH range this semisolid state applies. NaNO3 is certainly dissolved and
- 527 liquid-like in dilute aqueous aerosols at elevated RH (for aqueous NaNO3 viscosity data see
- 528 e.g. Baldelli et al., 2016, doi:10.1080/02786826.2016.1177163; Lilek and Zuend, 2022,
- 529 doi:10.5194/acp-22-3203-2022).
- Author reply: Thanks for the reviewer's suggestion. We have adopted the reviewer's adviceand revised our manuscript accordingly.
- 47. Line 427: "metastable or liquid-like state..." Please rephrase; metastable and liquid-like
  characterize two different properties. One related to the supersaturation of aqueous solutions
  with respect to a certain crystalline phase, while the other refers to a phase state (viscosity
  related).
- Author reply: Thanks for the reviewer's suggestion. We have adopted the reviewer's adviceand revised our manuscript accordingly.
- 538 Supplement: minor comments

539 48. SI, lines 20-36: How was that ERH determined? Was it based on just some features of the 540 spectra? Fig. S1a does not seem to indicate any clear signature of efflorescence. Specifically, 541 note that in the absence of severe kinetic limitations, one would expect a sharp change in mass 542 growth factor at the ERH (point of crystallization) of a single-solute particle. Certainly so for 543 a system at equilibrium with the gas phase. Why is that not the case for the systems shown in Fig. S1? Figure S2 shows also no indication of NaNO3 crystallization. Do the authors use the 544 term "efflorescence" for a phase transition other than crystallization? If so, that would be 545 546 untypical given the common use and meaning of this term and might need clarification in the 547 text.

548 Author reply: The ERH of NaNO3 was determined via the IR feature changes of NO3-, i.e., 549 the broad  $\sim 1350 \text{ cm}^{-1}$  band was transformed into a sharper peak, as shown in Fig. S1b and 550 S2b. The absence of sharp decrease in mass growth factor at the ERH was attributed to the 551 specific structure of nitrates after drying, i.e., amorphous solids. Likewise, Li et al. (2021) 552 observed the hygroscopic behaviours of pure  $(NH_4)$ 2SO4 and  $NH_4NO_3$  particles during the RH 553 cycles. They found that mass growth factor of  $(NH_4)_2SO_4$  particles would decrease sharply at the ERH, while mass growth factor of NH4NO3 particles decreased routinely without the 554 turning point at the ERH. Meanwhile, the 1435 cm-1 band assigned to  $NH_4^+$  in the solution 555 phase red-shifted to 1415 cm-1, suggesting the occurrence of efflorescence of NH4NO3 556 particles at 25% RH. Furthermore, Tang and Fung (1997) found that Raman spectra 557 558 characteristic of dried Ca(NO3)2 particles was obviously different from that of anhydrous 559 crystals, indicating the amorphous solids formation. Meanwhile, no sharp decrease in the water content of Ca(NO3)2 particles was observed during the dehumidification, and the 560 561 amorphous solids would deliquesce at RH well below the DRH of anhydrous crystals, which showed good agreement with the observations for NaNO3 particles in this work. In addition, 562 the term "efflorescence" did indicate the formation of amorphous solids rather than the 563 564 crystallization here. For clarity, we have revised our manuscript accordingly.

565 49. SI, line 40: what about that of a NaNO3 hydrate crystal? Would equilibrium thermodynamics
566 not require that amorphous solids take up some water upon hydration (while single-solute
567 crystalline particles may not)?

568 Author reply: As indicated in the Supplement, the  $v_3(NO_3)$  feature bands after drying in the two measurements are inconsistent with the IR feature of NaNO3 crystals, i.e., two shoulder 569 peaks centred at 1383 and 1485 cm-1 arising from splitting of the degenerate  $v_3$  mode (Liu et 570 571 al., 2008). Liu et al. (2008) also found small amounts of water present within particles after 572 dehydration and the particles absorb water continuously upon hydration, indicating the 573 formation of highly concentrated droplets after drying, rather than amorphous NaNO3 solids. 574 The amorphous nitrate solids (e.g.,  $Sr(NO_3)_2$  and  $Ca(NO_3)_2$ ) would not obviously absorb 575 water until their DRHs (Tang and Fung, 1997).

- 576 50. SI, lines 155–158, 166: these sentences are inconsistent. One is about HCl release, the other
  577 about HNO3 but referring to the same study. Please clarify.
- 578 Author reply: Thanks for the reviewer's suggestion. We have revised the error in the revised579 manuscript.
- 580 51. SI, line 178: molecular weight (or rather molar mass) of water should be stated in kg  $mol^{-1}$  to

stay consistent with the other units given. Other there will be an incorrect Kelvin effect value.

- 582 Author reply: Thanks for the reviewer's suggestion. We have revised the error in the revised
  583 manuscript.
- 584 52. SI, line 188: It should be stated that the true equilibria in aqueous solution involve use of the 585 activities of the species, not the molar concentrations. Also, on line 210, state that 586 self-dissociation of  $H_2O$  is not considered.
- 587 Author reply: Thanks for the reviewer's suggestion. The calculated pH here can be defined 588 as free-H+ approximation of pH on a molality basis, as indicated by Pye et al. (2020). 589 Furthermore, we have considered the self-dissociation of  $H_2O$  in eq. (11) in the revised 590 Supplement.
- 53. SI, line 216: given definition of pH; note that this refers only to an approximate,
  molarity-based pH, as discussed by Pye et al. (2020, doi:10.5194/acp-20-4809-2020). The
- 593 proper definition of pH involves the molality-based activity of  $H^+$ .
- Author reply: Thanks for the reviewer's suggestion. We have adopted the reviewer's adviceand revised our manuscript accordingly.
- 596

**597 Reference:**

- Bruzewicz, D. A., Checco, A., Ocko, B. M., Lewis, E. R., McGraw, R. L., and Schwartz, S. E.:
  Reversible uptake of water on NaCl nanoparticles at relative humidity below deliquescence point
  observed by noncontact environmental atomic force microscopy, J. Chem. Phys., 134, 044702,
  https://doi.org/10.1063/1.3524195, 2011.
- 602 Chen, Z., Liu, P., Liu, Y., and Zhang, Y. H.: Strong acids or bases displaced by weak acids or bases in
- aerosols: Reactions driven by the continuous partitioning of volatile products into the gas phase, Acc.
- 604 Chem. Res., 54, 3667-3678, https://doi.org/10.1021/acs.accounts.1c00318, 2021.
- Du, C. Y., Yang, H., Wang, N., Pang, S. F., and Zhang, Y. H.: pH effect on the release of NH3 from the
  internally mixed sodium succinate and ammonium sulfate aerosols, Atmos. Environ., 220, 117101,
  https://doi.org/10.1016/j.atmosenv.2019.117101, 2020.
- Esat, K., David, G., Poulkas, T., Shein, M., and Signorell, R.: Phase transition dynamics of single
  optically trapped aqueous potassium carbonate particles, Phys. Chem. Chem. Phys., 20, 11598-11607,
  https://doi.org/10.1039/c8cp00599k, 2018.
- 611 Ghorai, S., Wang, B. B., Tivanski, A., and Laskin, A.: Hygroscopic properties of internally mixed
- 612 particles composed of NaCl and water-soluble organic acids, Environ. Sci. Technol., 48, 2234-2241,
- 613 https://doi.org/10.1021/es404727u, 2014.
- 614 Gibson, E. R., Hudson, P. K., and Grassian, V. H.: Physicochemical properties of nitrate aerosols:
- 615 Implications for the atmosphere, J. Phys. Chem. A, 110, 11785-11799,

- 616 https://doi.org/10.1021/jp063821k, 2006.
- 617 He, X., Leng, C., Pang, S., and Zhang, Y.: Kinetics study of heterogeneous reactions of ozone with
- unsaturated fatty acid single droplets using micro-FTIR spectroscopy, RSC Adv., 7, 3204-3213,
  https://doi.org/10.1039/c6ra25255a, 2017.
- 620 Hoffman, R. C., Gebel, M. E., Fox, B. S., and Finlayson-Pitts, B. J.: Knudsen cell studies of the
- 621 reactions of N2O5 and ClONO2 with NaCl: development and application of a model for estimating
- available surface areas and corrected uptake coefficients, Phys. Chem. Chem. Phys., 5, 1780-1789,
- 623 https://doi.org/10.1039/B301126G, 2003a.
- 624 Hoffman, R. C., Kaleuati, M. A., and Finlayson-Pitts, B. J.: Knudsen cell studies of the reaction of
- gaseous HNO3 with NaCl using less than a single layer of particles at 298 K: A modified mechanism, J.
  Phys. Chem. A, 107, 7818-7826, https://doi.org/10.1021/jp0306110, 2003b.
- Hoffman, R. C., Laskin, A., and Finlayson-Pitts, B. J.: Sodium nitrate particles: physical and chemical
  properties during hydration and dehydration, and implications for aged sea salt aerosols, J. Aerosol Sci.,
  35, 869-887, https://doi.org/10.1016/j.jaerosci.2004.02.003, 2004.
- Ji, Z. R., Zhang, Y., Pang, S. F., and Zhang, Y. H.: Crystal nucleation and crystal growth and mass
  transfer in internally mixed sucrose/NaNO3 particles, J. Phys. Chem. A, 121, 7968-7975,
  https://doi.org/10.1021/acs.jpca.7b08004, 2017.
- 633 Jing, B., Wang, Z., Tan, F., Guo, Y. C., Tong, S. R., Wang, W. G., Zhang, Y. H., and Ge, M. F.:
- 634 Hygroscopic behavior of atmospheric aerosols containing nitrate salts and water-soluble organic acids,
- 635 Atmos. Chem. Phys., 18, 5115-5127, https://doi.org/10.5194/acp-18-5115-2018, 2018.
- 636 Kerminen, V.-M., Teinilä, K., Hillamo, R., and Pakkanen, T.: Substitution of chloride in sea-salt 637 particles by inorganic and organic anions, J. Aerosol Sci., 29, 929-942, 638 https://doi.org/10.1016/S0021-8502(98)00002-0, 1998.
- 639 Laskin, A., Moffet, R. C., Gilles, M. K., Fast, J. D., Zaveri, R. A., Wang, B. B., Nigge, P., and
- 640 Shutthanandan, J.: Tropospheric chemistry of internally mixed sea salt and organic particles: Surprising
- reactivity of NaCl with weak organic acids, J. Geophys. Res.-Atmos., 117, D15302,
  https://doi.org/10.1029/2012jd017743, 2012.
- 643 Li, Q., Ma, S. S., Pang, S. F., and Zhang, Y. H.: Measurement on mass growth factor of (NH4)2SO4,
- $M_4NO_3$  and mixed  $(NH_4)_2SO_4/NH_4NO_3$  aerosols under linear RH changing mode, Spectrosc. Spectral
- 645 Anal., 41, 3444-3450, https://doi.org/10.3964/j.issn.1000-0593(2021)11-3444-07, 2021.
- Li, X., Gupta, D., Lee, J., Park, G., and Ro, C.-U.: Real-time investigation of chemical compositions
- 647and hygroscopic properties of aerosols generated from NaCl and malonic acid mixture solutions using648in situ Raman microspectrometry, Environ. Sci. Technol., 51, 263-270,649100 1021/1
- 649 https://doi.org/10.1021/acs.est.6b04356, 2017.
- Liu, Y., Yang, Z. W., Desyaterik, Y., Gassman, P. L., Wang, H., and Laskin, A.: Hygroscopic behavior
   of substrate-deposited particles studied by micro-FT-IR spectroscopy and complementary methods of
- 652 particle analysis, Anal. Chem., 80, 633-642, https://doi.org/10.1021/ac701638r, 2008.
- Ma, Q. X., Ma, J. Z., Liu, C., Lai, C. Y., and He, H.: Laboratory study on the hygroscopic behavior of
- 654 external and internal  $C_2$ - $C_4$  dicarboxylic acid-NaCl mixtures, Environ. Sci. Technol., 47, 10381-10388, 655 https://doi.org/10.1021/es4023267, 2013.
- 656 Ma, Q. X., Zhong, C., Liu, C., Liu, J., Ma, J. Z., Wu, L. Y., and He, H.: A comprehensive study about
- 657 the hygroscopic behavior of mixtures of oxalic acid and nitrate salts: Implication for the occurrence of
- 658 atmospheric metal oxalate complex, ACS Earth Space Chem., 3, 1216-1225,
- 659 https://doi.org/10.1021/acsearthspacechem.9b00077, 2019a.

- 660 Ma, S. S., Yang, W., Zheng, C. M., Pang, S. F., and Zhang, Y. H.: Subsecond measurements on aerosols: 661 From hygroscopic growth factors to efflorescence kinetics, Atmos. Environ., 210, 177-185, 662 https://doi.org/10.1016/j.atmosenv.2019.04.049, 2019b.
- 663 Ma, S. S., Yang, M., Pang, S. F., and Zhang, Y. H.: Subsecond measurement on deliquescence kinetics 664 of aerosol particles: Observation of partial dissolution and calculation of dissolution rates, 665 Chemosphere, 264, 128507, https://doi.org/10.1016/j.chemosphere.2020.128507, 2021.
- Pye, H. O. T., Nenes, A., Alexander, B., Ault, A. P., Barth, M. C., Clegg, S. L., Collett Jr, J. L., Fahey, 666
- 667 K. M., Hennigan, C. J., Herrmann, H., Kanakidou, M., Kelly, J. T., Ku, I. T., McNeill, V. F., Riemer, N.,
- 668 Schaefer, T., Shi, G., Tilgner, A., Walker, J. T., Wang, T., Weber, R., Xing, J., Zaveri, R. A., and Zuend,
- 669 A.: The acidity of atmospheric particles and clouds, Atmos. Chem. Phys., 20, 4809-4888, 670 https://doi.org/10.5194/acp-20-4809-2020, 2020.
- 671 Ren, H. M., Cai, C., Leng, C. B., Pang, S. F., and Zhang, Y. H.: Nucleation kinetics in mixed 672 NaNO3/glycerol droplets investigated with the FTIR-ATR technique, J. Phys. Chem. B, 120, 673 2913-2920, https://doi.org/10.1021/acs.jpcb.5b12442, 2016.
- 674 Shao, X., Wu, F. M., Yang, H., Pang, S. F., and Zhang, Y. H.: Observing HNO3 release dependent upon metal complexes in malonic acid/nitrate droplets, Spectrochim. Acta A, 201, 399-404, 675 676 https://doi.org/10.1016/j.saa.2018.05.026, 2018.
- 677 Tang, I. N., and Fung, K. H.: Hydration and Raman scattering studies of levitated microparticles: 678 Ba(NO3)2, Sr(NO3)2, and Ca(NO3)2, J. Chem. Phys., 106, 1653-1660, https://doi.org/10.1063/1.473318, 679 1997.
- 680 Wang, B. B., and Laskin, A.: Reactions between water-soluble organic acids and nitrates in 681 atmospheric aerosols: Recycling of nitric acid and formation of organic salts, J. Geophys. Res.-Atmos., 682 119, 3335-3351, https://doi.org/10.1002/2013jd021169, 2014.
- 683 Wang, N., Jing, B., Wang, P., Wang, Z., Li, J. R., Pang, S. F., Zhang, Y. H., and Ge, M. F.: 684 Hygroscopicity and compositional evolution of atmospheric aerosols containing water-soluble 685 carboxylic acid salts and ammonium sulfate: Influence of ammonium depletion, Environ. Sci. Technol., 686 53, 6225-6234, https://doi.org/10.1021/acs.est.8b07052, 2019.
- Wang, X. W., Jing, B., Tan, F., Ma, J. B., Zhang, Y. H., and Ge, M. F.: Hygroscopic behavior and 687 688 chemical composition evolution of internally mixed aerosols composed of oxalic acid and ammonium 689 sulfate, Atmos. Chem. Phys., 17, 12797-12812, https://doi.org/10.5194/acp-17-12797-2017, 2017.
- 690 Wise, M. E., Martin, S. T., Russell, L. M., and Buseck, P. R.: Water uptake by NaCl particles prior to
- 691 deliquescence and 281-294, the phase rule, Aerosol Sci. Technol., 42, 692 https://doi.org/10.1080/02786820802047115, 2008.
- Wu, Z. J., Nowak, A., Poulain, L., Herrmann, H., and Wiedensohler, A.: Hygroscopic behavior of 693
- 694 atmospherically relevant water-soluble carboxylic salts and their influence on the water uptake of
- 695 ammonium sulfate, Atmos. Chem. Phys., 11, 12617-12626, https://doi.org/10.5194/acp-11-12617-2011, 2011.
- 696
- 697 Zhang, Q. N., Zhang, Y., Cai, C., Guo, Y. C., Reid, J. P., and Zhang, Y. H.: In situ observation on the
- 698 dynamic process of evaporation and crystallization of sodium nitrate droplets on a ZnSe substrate by
- 699 FTIR-ATR, J. Phys. Chem. A, 118, 2728-2737, https://doi.org/10.1021/jp412073c, 2014.

---

## Author Response (AR2)

**Response to Reviewers:**

Thanks for the reviewer's comments on our manuscript entitled " A comprehensive study on hygroscopic behaviour and nitrate depletion of $NaNO_3$ and dicarboxylic acid mixtures: Implications for nitrate depletion in tropospheric aerosols". The reviewers' comments are helpful for improving the quality of our work. The responses to the comments and the revisions in manuscript are given point-to-point below.

**Comments:**

1.  Line 140: Sentence "These scenarios confirm the nitrate depletion and HNO3 release from NaNO3/OA mixtures in the vacuuming process."

    Consider rephrasing to clarify that this nitric acid release is assumed to take place in the present study's experiments, while it is not directly observed (the major comment from the first round of reviews). For example, wording like:

    "These observations, together with direct and indirect evidence from several past studies, confirm the release of HNO3 and associated nitrate depletion from NaNO3/OA aerosol particles, which is expected to occur during the vacuuming process employed."

    **Author reply:** Thanks for the reviewer's suggestion. We have adopted reviewer's advice and revised our manuscript accordingly. The sentence "These scenarios confirm the nitrate depletion and $HNO_3$ release from $NaNO_3$/OA mixtures in the vacuuming process." has been revised to "As already indicated, the release of $HNO_3$ and associated organic acid salts formation have been detected in several previous studies, thus herein, these observations can demonstrate the $HNO_3$ release and nitrate depletion in $NaNO_3$/OA mixtures, which is expected to occur in the vacuuming process.".

2.  Line 315: revise the second sentence: "There are two probable causes for no chloride depletion observed in the fast drying process. One is the minimization of HNO3 release caused by rapid water evaporation (Ma et al., 2013)."

    This is about chloride depletion, so presumably it should be HCl not HNO3 in the second sentence.

    **Author reply:** Thanks for the reviewer's suggestion. We have adopted reviewer's advice and revised our manuscript accordingly.

3.  Line 453: the added sentence requires revision: "In atmospheric environment, insoluble materials such as mineral dust inclusions constantly induce the heterogeneous nucleation of aerosols at relatively high RH, and thus displacement reactions between MA or GA and nitrate rarely contribute to the nitrate depletion in mineral dust and sea salt aerosols."

I am unsure what you mean by "heterogeneous nucleation of aerosols" in this context (do you rather mean nucleation of solid salt phases?). Obviously, if you have mineral dust inclusions, there is no need for "aerosol nucleation". Also, the terms "constantly" and "nucleation" are contradictory; nucleation is a discrete, event-based process (perhaps you mean frequently instead of constantly).

**Author reply:** Thanks for the reviewer's suggestion. The "heterogeneous nucleation of aerosols" did indicate the crystallization of mixed droplets in the atmosphere, which contained organic and inorganic components and small amounts of mineral dust inclusions. The insoluble mineral dust inclusions could provide heterogeneous surfaces and induce the heterogeneous nucleation of atmospheric aerosols at relatively high RH (Ma et al., 2021). For clarity, we have revised the sentence "In atmospheric environment, insoluble materials such as mineral dust inclusions constantly induce the heterogeneous nucleation of aerosols at relatively high RH, and thus displacement reactions between MA or GA and nitrate rarely contribute to the nitrate depletion in mineral dust and sea salt aerosols." into "In atmospheric aerosols, insoluble materials such as mineral dust inclusions frequently induce the heterogeneous nucleation of aerosol droplets at relatively high RH, and thus displacement reactions between MA or GA and nitrates may rarely contribute to the nitrate depletion in aerosols.".

Reference:

Ma, S. S., Pang, S. F., Li, J., and Zhang, Y. H.: A review of efflorescence kinetics studies on atmospherically relevant particles, Chemosphere, 277, 130320, https://doi.org/10.1016/j.chemosphere.2021.130320, 2021.